# Discovery of methylfarnesoate as the annelid brain hormone reveals an ancient role of sesquiterpenoids in reproduction

Sven Schenk[1,2]*, Christian Krauditsch[1], Peter Frühauf[2,3], Christopher Gerner[2,3], Florian Raible[1,2]*

[1]Max F. Perutz Laboratories, University of Vienna, Vienna Biocenter (VBC), Vienna, Austria; [2]Research Platform Marine Rhythms of Life, University of Vienna, Vienna Biocenter (VBC), Vienna, Austria; [3]Institute for Analytical Chemistry, University of Vienna, Vienna, Austria

**Abstract** Animals require molecular signals to determine when to divert resources from somatic functions to reproduction. This decision is vital in animals that reproduce in an all-or-nothing mode, such as bristle worms: females committed to reproduction spend roughly half their body mass for yolk and egg production; following mass spawning, the parents die. An enigmatic brain hormone activity suppresses reproduction. We now identify this hormone as the sesquiterpenoid methylfarnesoate. Methylfarnesoate suppresses transcript levels of the yolk precursor Vitellogenin both in cell culture and *in vivo*, directly inhibiting a central energy–costly step of reproductive maturation. We reveal that contrary to common assumptions, sesquiterpenoids are ancient animal hormones present in marine and terrestrial lophotrochozoans. In turn, insecticides targeting this pathway suppress vitellogenesis in cultured worm cells. These findings challenge current views of animal hormone evolution, and indicate that non-target species and marine ecosystems are susceptible to commonly used insect larvicides.

*For correspondence: sven.schenk@univie.ac.at (SS); florian.raible@univie.ac.at (FR)

**Competing interests:** The authors declare that no competing interests exist.

## Introduction

As animals rely on limited energy resources, they require regulatory mechanisms to decide how to best invest available energy and biomaterial. Some of the most extreme changes of energy expenditure are observed in so-called semelparous animals that reproduce only once and typically die afterwards. In both vertebrate and invertebrate representatives of semelparous animals, the switch to the reproductive state is accompanied by a massive breakdown of somatic tissue, in favor of forming and nourishing germ cells: migrating salmon break down about 50% of their white muscle mass while their gonad mass multiplies around six-fold during maturation (*Mommsen, 2004*). Likewise, female squids exhibit almost complete histolysis of mantle muscle (*Jackson and Mladenov, 1994*), and in mature female squids up to ~60% of the total body mass at spawning consists of eggs (*Rodrigues et al., 2011*). Similarly, in the marine annelid worm *Nereis virens*, oocytes represent up to about 40% of the body mass of the spawning adult (*Hoeger et al., 1999*).

Insight into the molecules regulating these extreme shifts of energy expenditure has remained limited (*Cardon et al., 1981*; *Mommsen, 2004*; *Hébert Chatelain et al., 2008*). This can largely be attributed to the lack of an appropriate laboratory model system amenable to molecular research. In this study, we explore the marine annelid *Platynereis dumerilii*, a semelparous animal that can easily be reared in the laboratory (*Hauenschild and Fischer, 1969*; *Fischer and Dorresteijn, 2004*) and has been established as a molecular model system over the last decade (reviewed in *Zantke et al., 2014*). A series of studies show that *Platynereis* has retained various ancient-type characteristics

**eLife digest** All organisms need energy to survive and grow. However, sources of energy are limited and so organisms need to decide how to spend the resources they have available. For instance, animals must choose whether they should continue to grow or if they should invest energy into reproduction instead. This decision becomes even more important for animals that reproduce in an "all-or-nothing" manner and invest all their available energy into reproduction and die soon after.

Bristle worms live in coastal areas around world. In mass spawning events, thousands of individuals raise from the sea floor to the surface, to release sperm and eggs. While the fertilized eggs start to develop in the water, the parents invariably die. The female worms spend roughly half their body mass in producing eggs and supplying them with yolk as a source of energy. It has been known for decades that the brains of bristle worms produce a master hormone that promotes growth and suppresses reproduction. Yet the identity of this hormone that controls the life-or-death decision was not clear.

Schenk et al. took advantage of new molecular tools to solve this puzzle. The experiments show that this hormone directly regulates how much yolk the female animals produce. This allowed Schenk et al. to design a new molecular assay that helped to identify the hormone itself. Unexpectedly, the hormone – called methylfarnesoate – belongs to a family of small molecules called sesquiterpenoids, which researchers previously thought were only found in insects and related groups. Hence, many insecticides have been developed to target sesquiterpenoid signaling and they are used in massive amounts to fight pests like the tiger mosquito (which transmits the Zika virus). Schenk et al. also found that these insecticides also cause severe problems in bristle-worms.

These findings challenge current views of how animal hormones have evolved and indicate that common insecticides may be harming bristle worms and other animals in marine environments. The next steps are to find out whether methylfarnesoate is found in other closely related animals, such as snails and mussels, and whether the insecticides are harmful to these animals too. Another future challenge will be to investigate how this hormone actually promotes animal growth.

---

(*Arendt et al., 2004*; *Raible et al., 2005*; *Denes et al., 2007*; *Tessmar-Raible et al., 2007*; *Keay and Thornton, 2009*; *Christodoulou et al., 2010*), compatible with the notion that the hormones present in this species are also more broadly comparable with other clades.

In *Platynereis*, the critical switch to the reproductive life stage has been linked to a hormonal activity released by the medial brain. This brain hormonal activity – also termed nereidin – was found to repress sexual maturation and to promote growth of the trunk and posterior regeneration (*Figure 1A*): Decerebration of *Platynereis* leads to an accelerated sexual maturation and to the loss of regenerative abilities, whereas re-implantation of a juvenile head with neuroendocrine activity is able to reverse these effects (*Hauenschild, 1956a*, *1956b*, *1960*, *1966*). In the related species *Nereis diversicolor*, it could be shown that implantation of ganglia from immature animals into the coelomic cavity of maturing animals, reversed the effects of the decreasing nereidin concentration. In particular, germ cells were resorbed and the ability to regenerate lost segments was re-established (*Golding and Yuwono, 1994*). These findings highlight the exceptional importance of neuroendocrine signals in semelparous reproduction and suggest that the underlying molecular mechanisms are conserved across different annelids (*Hauenschild, 1956b*, *1959*; *Golding, 1967a*, *1967b*; *Durchon and Porchet, 1970*; *Hofmann, 1976*; *Cardon et al., 1981*; *Golding, 1983*; *Golding and Yuwono, 1994*).

Although nereidin activity was recognised as a critical cue more than 60 years ago, the molecular identity of nereidin still remains elusive. However, the results of classical experiments allow for several hypotheses concerning the function and molecular properties of this key brain hormone: (i) When present, nereidin interferes with a critical step required for the animals to enter the cost-intensive process of maturation. (ii) The effective hormone concentration decreases from young animals to mature animals, such that the drop in nereidin concentration allows maturation to proceed (*Hauenschild, 1956a*, *1956b*, *1966*; *Golding, 1983*). (iii) As Nereidin-like activities have been identified in various annelids, this suggests that these substances could be chemically related.

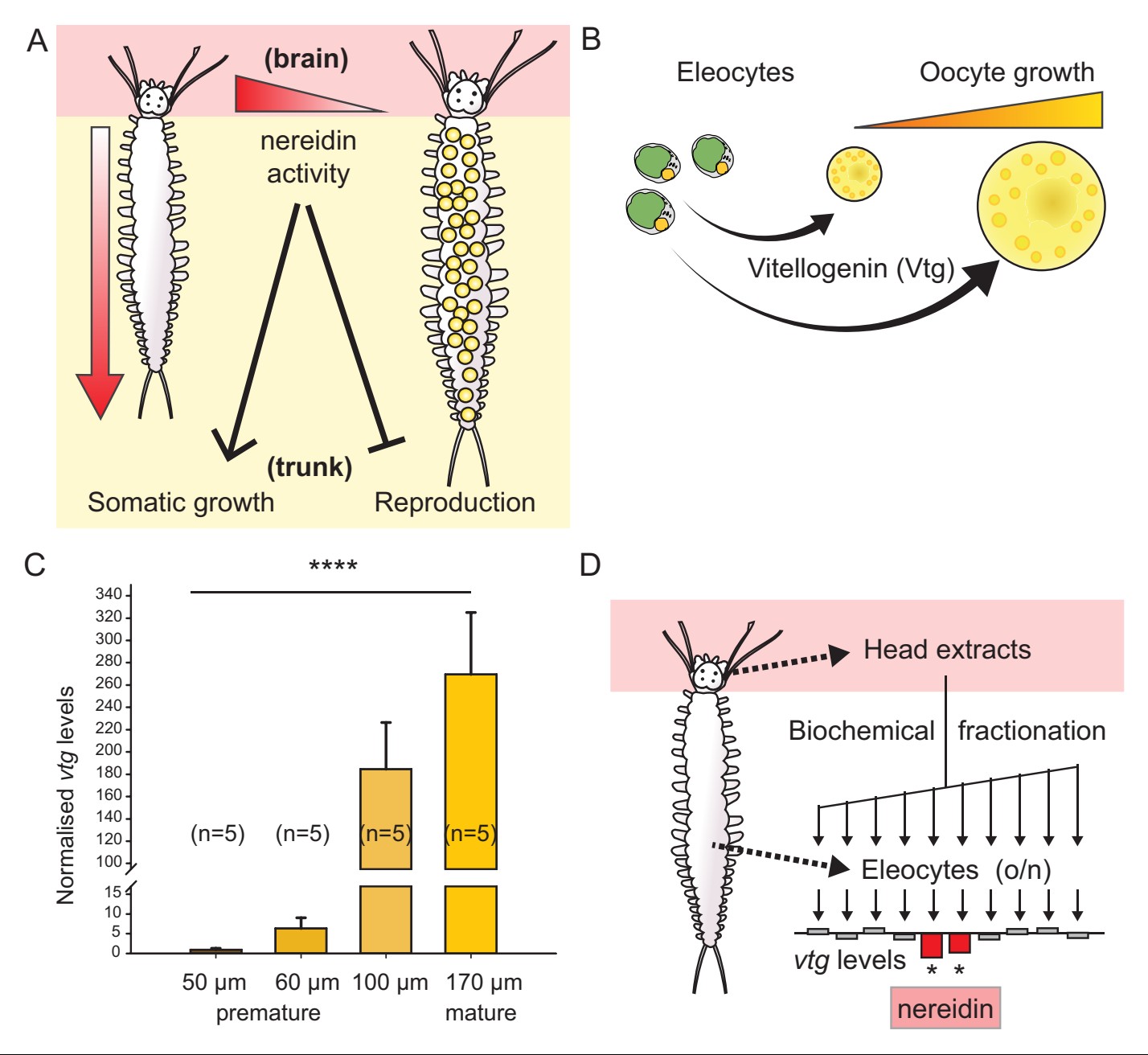

**Figure 1.** *Vitellogenin* expression in eleocytes is a quantitative measure for the maturation stage of *Platynereis*, allowing for the establishment of a bioassay to purify the enigmatic brain hormone, nereidin. (A) Scheme summarising the critical role of the brain hormone nereidin in energy expenditure, as derived from classical experiments: Before maturation (left), high nereidin levels sustain somatic growth, but repress reproduction; drops in nereidin activity levels initiate the generation of germ cells and sexual maturation (right). (B) Eleocytes in the worm's coelom have a central role in reproductive commitment, as they synthesise the yolk protein precursor Vitellogenin (Vtg); after release into the coelomic fluid, Vtg is endocytosed by the developing oocytes, leading to oocyte growth. (C) Expression levels of *vtg* in eleocytes are significantly up-regulated during maturation, confirming *vtg* as a suitable molecular marker to quantify maturation state. qRT-PCR quantification of *vtg* levels in eleocytes sampled at different stages of sexual maturation (as assessed by oocyte diameter). The increase in *vtg* transcripts from premature eleocytes (oocyte diameter: 50 and 60 μm) to those from mature eleocytes (oocytes diameter: 170 μm) is evident; expression levels were calculated against the arithmetic mean of the reference genes *cdc5* and *rps9*, and normalised to the expression of *vtg* in eleocytes from animals with an oocyte diameter of 50 μm. Statistical significance was tested an one-way ANOVA. ***$p < 0.001$. n: number of biological replicates. (D) Resulting bioassay for the purification of nereidin from head extracts: primary cell cultures were derived from coelomic cells of premature animals; *vtg* levels were quantified after overnight incubation with different fractions of head extract to determine those fractions containing significant inhibitory activity (nereidin). Data for panel (C) provided in *Figure 1—source data 1*.

*Figure 1 continued on next page*

*Figure 1 continued*

The following source data and figure supplements are available for figure 1:

**Source data 1.** Data for the graphs in *Figure 1C*, *Figure 1—figure supplement 2A*, *Figure 1—figure supplement 2B* (*vtg* expression levels over the time course of maturation).

**Source data 2.** Alignment file for the phylogenetic tree of Vitellogenins (*Figure 1—figure supplement 1B*).

**Figure supplement 1.** Identification of a *Platynereis* Vitellogenin orthologue.

**Figure supplement 2.** Irrespective of the chosen reference gene, *vitellogenin* is regulated over the course of maturation.

Histological analyses of the proposed biosynthetically active brain region (*Müller, 1973*; *Hofmann, 1976*) were compatible with a fourth hypothesis, namely that nereidin is a neuropeptide. Based on this hypothesis, large-scale purifications have been performed from head extracts of *N. diversicolor* and *Perinereis cultrifera* (*Cardon et al., 1981*). These finally resulted in the identification of a hexapeptide (PPGPPG). However, this peptide did not display biological activity (*Durchon, 1984*).

In this study, we take advantage of a newly developed, quantitative cell culture-based bioassay that measures transcript levels of the main *Platynereis* yolk precursor Vitellogenin (Vtg). Using this assay, we demonstrate that nereidin activity is not sensitive to proteinase treatment, challenging the long-standing hypothesis that nereidin is a peptide. Instead, we correlate nereidin activity with methylfarnesoate (MF), a substance belonging to the hormone class of sesquiterpenoids that has so far been assumed to be restricted to arthropods. We demonstrate that the concentration of MF drops in the course of maturation – as predicted from classical nereidin activity assays – and that eleocytes express the orthologue of a sesquiterpenoid receptor. This, together with our finding that MF exerts its action on *vitellogenin* production not only in cell culture, but also *in vivo*, establishes MF as a brain hormone orchestrating a critical metabolic switch in *P. dumerilii*. We finally show that insecticides targeting the sesquiterpenoid pathway interfere with *Platynereis* vitellogenesis, revealing a profound impact of insect growth regulators on a presumed non-target organism that serves as a key marine reference species.

## Results

### A fast and sensitive assay to assess maturation in *Platynereis*

One of the major challenges faced by previous attempts to identify the brain hormone activity was the absence of a fast and sensitive bioassay to measure the inhibitory effect of nereidin on sexual maturation. Fractionated head extracts could be shown to interfere with the transition of cultured spermatogonia to spermatozoa (*Cardon, 1970*; *Durchon and Porchet, 1970*; *Cardon et al., 1981*). However, the respective assay required more than a week of culturing and substantial input material (thousands of sampled heads). Using this qualitative assay, and systematic fractionating of head extracts, the brain hormone activity could be associated with a small ($M_R \leq 2.000$), lipophilic molecule (*Cardon, 1970*; *Durchon and Porchet, 1970*; *Cardon et al., 1981*).

In order to establish a sensitive read-out for *Platynereis* maturation, we decided to focus on the yolk protein precursor Vtg. Yolk production underlies the significant increase in oocyte diameter observed in nereidids during maturation (*Fischer, 1984*; *Fischer and Hoeger, 1993*). Yolk protein is derived by endocytotic uptake of Vtg into the oocyte (*Fischer et al., 1991*), where it accounts for more than 40% of protein (*Fischer and Schmitz, 1981*), reflecting its central role as reproductive investment. Vtg is synthesised in coelomic cells called eleocytes (*Figure 1B*) that are present in large numbers (*Baert and Slomianny, 1987*; *Hoeger, 1991*). Eleocytes and oocytes are freely floating in the coelomic cavity and can therefore easily be isolated and taken into primary (co)-culture, making them an excellent experimental model system to assess and monitor maturation in *Platynereis* and other nereidids, and to investigate the concomitant metabolic changes associated with this process.

Based on the central role of eleocytes and Vtg for oocyte growth, we hypothesised that the expression level of *vtg* in eleocytes might be a suitable molecular read-out for maturation. As a first step, we identified a *Platynereis vtg* candidate in eleocyte transcriptome data (Sven Schenk, Fritz Sedlazeck, Florian Raible, unpublished). Domain analyses and molecular phylogeny indicate that the protein encoded by this gene is an orthologue of Vtg identified in other animals (insects, molluscs, amphibians and birds; *Figure 1—figure supplement 1*). Next, we designed qRT-PCR primers to test if *vtg* transcript levels in *P. dumerilii* eleocytes increased over female maturation, as suggested by the increase of Vtg protein secretion in *P. cultrifera* (*Baert and Slomianny, 1992*). Indeed, we found a strong correlation between *vtg*-transcript abundance in eleocytes and the maturation state of the animal (as assessed by oocyte diameter, *Figure 1C* and *Figure 1—figure supplement 2*), indicating that *vtg* transcript levels could be used for a quantitative maturation assay. Given the central metabolic role of eleocytes in the reproductive switch, we further hypothesised that nereidin acts directly on eleocytes to regulate *vtg* production. We therefore designed a bioassay that takes coelomic cells into primary cell culture. After adding fractions of our purification scheme to small aliquots of this culture, and incubating these overnight, we extracted RNA from the cells and analysed them by qRT-PCR to determine the levels of *vtg* (*Figure 1D*).

## Methylfarnesoate is the active component of the brain hormone and acts at physiological concentrations

After establishing this bioassay, we used it to chemically isolate the brain hormone-activity of *Platynereis*. Similar to the method of *Cardon (1970)*, a methanolic extract was prepared from 50 heads of premature animals, i.e. animals becoming sexually distinct, but before the irreversible onset of maturation. After solvent evaporation, the material was reconstituted in an aqueous phase and further fractionated via phenyl-solid phase extraction (SPE, *Figure 2A*). Consistent with previous characterisations of nereidin, the 50% methanol eluate of this column showed clear activity (*Figure 2B*) when added to coelomocytes in vitro.

As outlined, it has long been speculated that nereidin is a peptide. This assumption was mainly based on the results of partial characterisation (*Durchon, 1984*), as well as histological observations, by which the brain regions producing nereidin were shown to contain highly active secretory cells (*Müller, 1973*; *Hofmann, 1976*). To assess whether the recovered nereidin activity was of proteinaceous nature, we treated the obtained eluate with proteinase K, and tested if this affected the ability of the eluate to repress *vtg* levels in cell culture. Our results demonstrate that nereidin is insensitive to proteinase K-treatment, thus strongly arguing against the notion that it is a neuropeptide hormone (*Figure 2C*).

To further characterise the recovered eluate, 250 premature worm heads were extracted and fractionated on a Phenyl-SPE-column. The recovered active fraction was then further fractionated by HPLC on a C18-column (*Figure 2D*). A total of 16 fractions eluting from the column were collected and tested in the cell culture-based bioassay for their impact on *vtg* transcript levels. One peak (peak 15) could be identified that robustly and significantly reduced *vtg* transcripts in two independent experiments and showed a reasonable chromatographic purity (*Figure 2D,E*). Analysis of peak 15 by LC-MS/MS clearly demonstrated the presence of the sesquiterpenoid methylfarnesoate (MF), when compared to an authentic standard run under the same conditions (*Figure 2—figure supplement 1*).

To test if MF was indeed the active compound in peak 15, we tested the effect of authentic MF on *vtg* transcript levels in the established cell culture assay. The results showed a clear down-regulation of *vtg* transcript levels (*Figure 2F*). Strongest effects were observed in the range between 1 and 100 nM MF, in agreement with physiological concentrations expected for a hormone. To ascertain the specificity of MF in down regulation of *vtg* transcripts in vitro, coelomic cells were also challenged with structurally related compounds such as palmitic acid (PA) and all-*trans*-retinoic acid (RA). As evident from *Figure 2G*, neither of these molecules showed a significant effect on vitellogenesis, thus establishing the specificity of MF on the observed reduction of *vtg* transcript levels.

## MF concentration drops in the course of maturation

Based on the classical brain transplantation experiments, it was hypothesised that nereidin is present at high levels in young animals, whereas its concentration declines during maturation

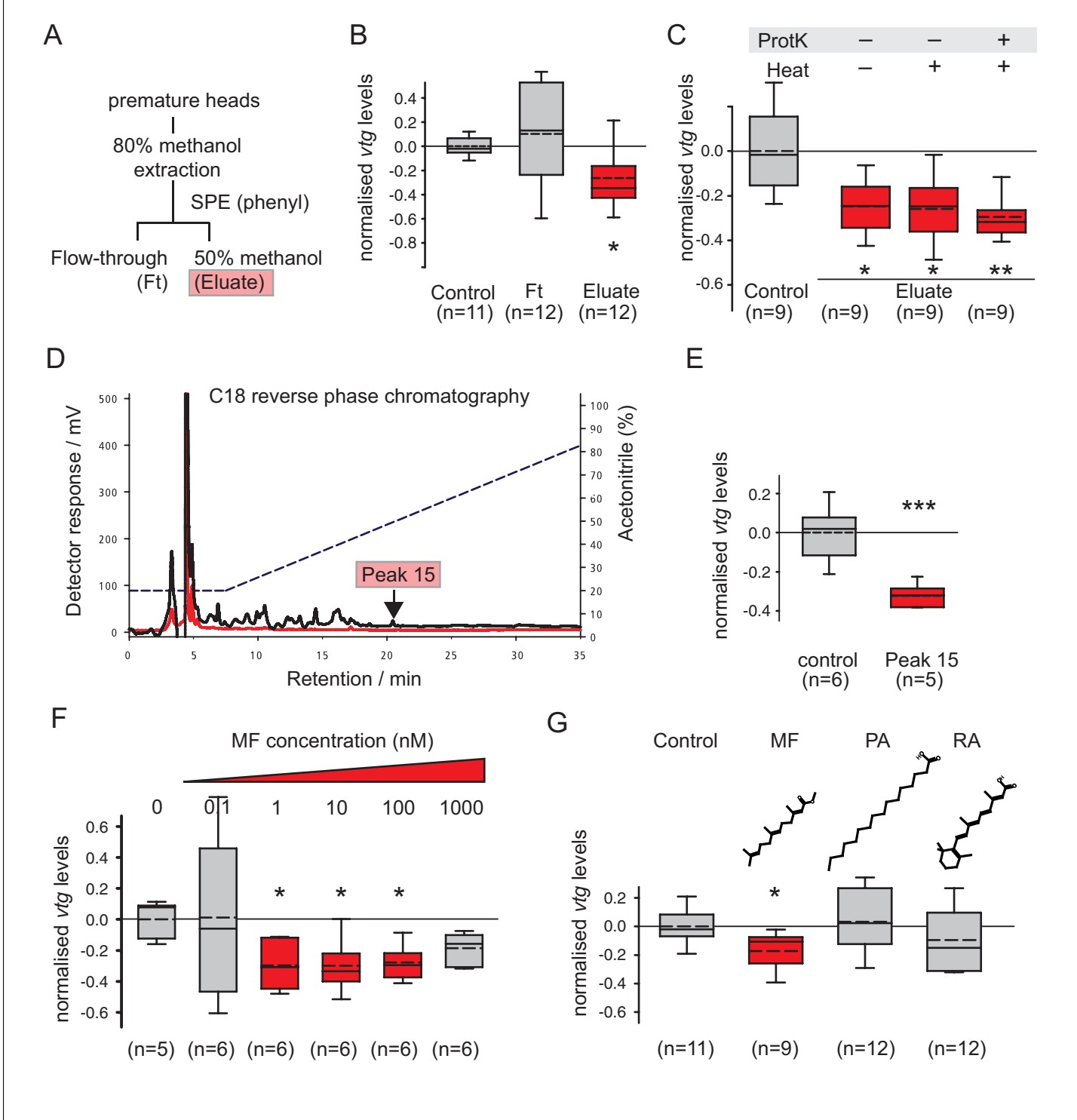

**Figure 2.** Methylfarnesoate, not protein, constitutes the main nereidin/brain hormone activity. (A–C) Nereidin activity elutes in lipophilic fractions of head extracts but is not proteinaceous as previously hypothesised. (A) Schematised fractionation of premature *Platynereis* head extracts, expected to recover the described nereidin activity in the methanolic eluate of SPE. (B) Relative coelomocyte *vtg* levels after overnight incubation with flowthrough or eluate of (a), normalised against control samples. (C) Insensitivity of nereidin activity against proteinase and heat treatment. Experiment as in (b), with pretreatment of eluate by proteinase K ('ProtK') and/or subsequent heat treatment (5', 95°C; 'Heat'). Results indicate that the repressive effect of nereidin on eleocyte *vtg* levels is neither sensitive to proteinase nor heat. (D–G) Identification of methylfarnesoate as a major component of nereidin. (D) Further fractioning of the brain hormone activity (eluate from b) by C18-reverse phase column chromatography. Absorption of eluent at 280 nm (red line) and 215 nm (black line) during application of acetonitrile gradient (dashed line, right ordinate). Peak 15 (marked red), containing the nereidin

*Figure 2 continued on next page*

*Figure 2 continued*

activity, elutes at a retention time of 20.5 min. (**E**) Peak 15, containing methylfarnesoate (MF), significantly suppresses *vtg* levels in the coelomocyte bioassay when compared to the controls. (**F**) Authentic MF recapitulates the observed effect at physiological concentration ranges; relative *vtg* levels in coelomocytes treated with 0.1 nM – 1000 nM MF in 0.01% DMSO, normalised to the control group of DMSO-treated cells. (**G**) Specificity of the repressive effect caused by MF; comparison of coelomocyte *vtg* levels after incubation in 10 nM MF, palmitic acid (PA) or retinoic acid (RA). Graphs in (**B,C,E,F,G**) show qRT-PCR quantification of *vtg* levels after overnight (20 hr) treatment of primary coelomocyte cultures. Expressions values were calculated with respect to the reference gene *rps9*, and normalised to the levels of controls (untreated cells). Boxplots show the first and third quartile, the median (solid line), and the mean (dashed line). Whiskers denote the 10th and 90th percentile. Statistical significance was tested by a one-sided t-test of the treatment group against the control, or, in case of multiple comparisons, first with an ANOVA, followed by a one-sided t-test with adjustment of the p-values for multiple testing to assess differences between the different groups. *$p<0.05$. **$p<0.01$, ***$p<0.001$. n: number of biological replicates. Raw data for panels B,C,E,F,G provided in *Figure 2—source data 1*.

The following source data and figure supplements are available for figure 2:

**Source data 1.** Data for *Figure 2B,C,E,F,G* (*vtg* expression levels normalised to the respective control treatment) and *Figure 2—figure supplement 2* (stability of Ct values for rps9).

**Figure supplement 1.** Identification of MF as the principal component in peak 15.

**Figure supplement 2.** Stability of the reference gene *rps9* in coelomocyte culture.

(*Hauenschild, 1956a*, *1956b*, *1966*; *Durchon and Porchet, 1970*; *Golding, 1983*). To test whether MF levels decrease over the course of maturation, we next set up a gas chromatography mass spectrometry (GC-MS)-based quantification assay.

Our extraction method followed the method described by *Westerlund and Hoffmann (2004)*, where a methanolic extract is used in combination with a phase partition against an organic solvent to simultaneously denature degrading enzymes and to extract MF. However, as we used heads to extract MF rather than haemolymph, we used in a first step 80% methanol (as in our initial extraction protocol, see above) followed by liquid-liquid extraction into heptane, instead of directly diluting the sample with the methanolic-organic extraction solution (*Westerlund and Hoffmann, 2004*). The extraction efficiency of our method was determined to be >85%. With an estimated limit of detection of ~2.5 pg MF and a limit of quantification of ~15 pg MF, our assay is comparable to those used by other authors for the identification of MF and related compounds (*Westerlund and Hoffmann, 2004*; *Teal et al., 2014*).

This assay allowed us to unequivocally identify MF in *Platynereis* heads (*Figure 3—figure supplement 1*) and to quantify the amount of MF by injecting 2.5 head equivalents into the GC-MS. Measuring MF levels present in heads of different developmental stages showed differences in MF content (*Figure 3A*). Heads of premature animals were determined to possess ~5510 ± 660 pg MF/mg protein (mean±standard error of the mean), whereas concentrations in the heads of mature animals were found to be significantly lower (~2870 ± 495 pg MF/mg protein). Calculated per head, the MF content is equivalent to ~34.6 ± 8.0 pg/head for premature heads and ~15.4 ± 2.5 pg/head for mature animals (*Figure 3—figure supplement 2*). Our results are therefore in agreement with the proposed decline of brain hormone concentrations during maturation.

In summary, the results of our bioassay and purification scheme demonstrate that the brain hormone-activity present in the heads of *P. dumerilii* is not a peptide. Instead, we clearly identify MF as the major molecule in *Platynereis* heads that is able to suppress transcription of *vtg* in coelomocyte cultures, when added at physiological concentrations. Furthermore, MF concentration drops in the transition from the premature animal to a mature stage, in agreement with classical predictions on the regulation of nereidin.

## *Platynereis* eleocytes possess an orthologue of validated arthropod sesquiterpenoid hormone receptors

The occurrence of MF in the annelid *Platynereis* is counterintuitive, because sesquiterpenoid hormones have commonly been proposed to be restricted to arthropods, possibly as a secondary consequence of the reduction of a parallel biosynthetic pathway leading to the formation of sterols in

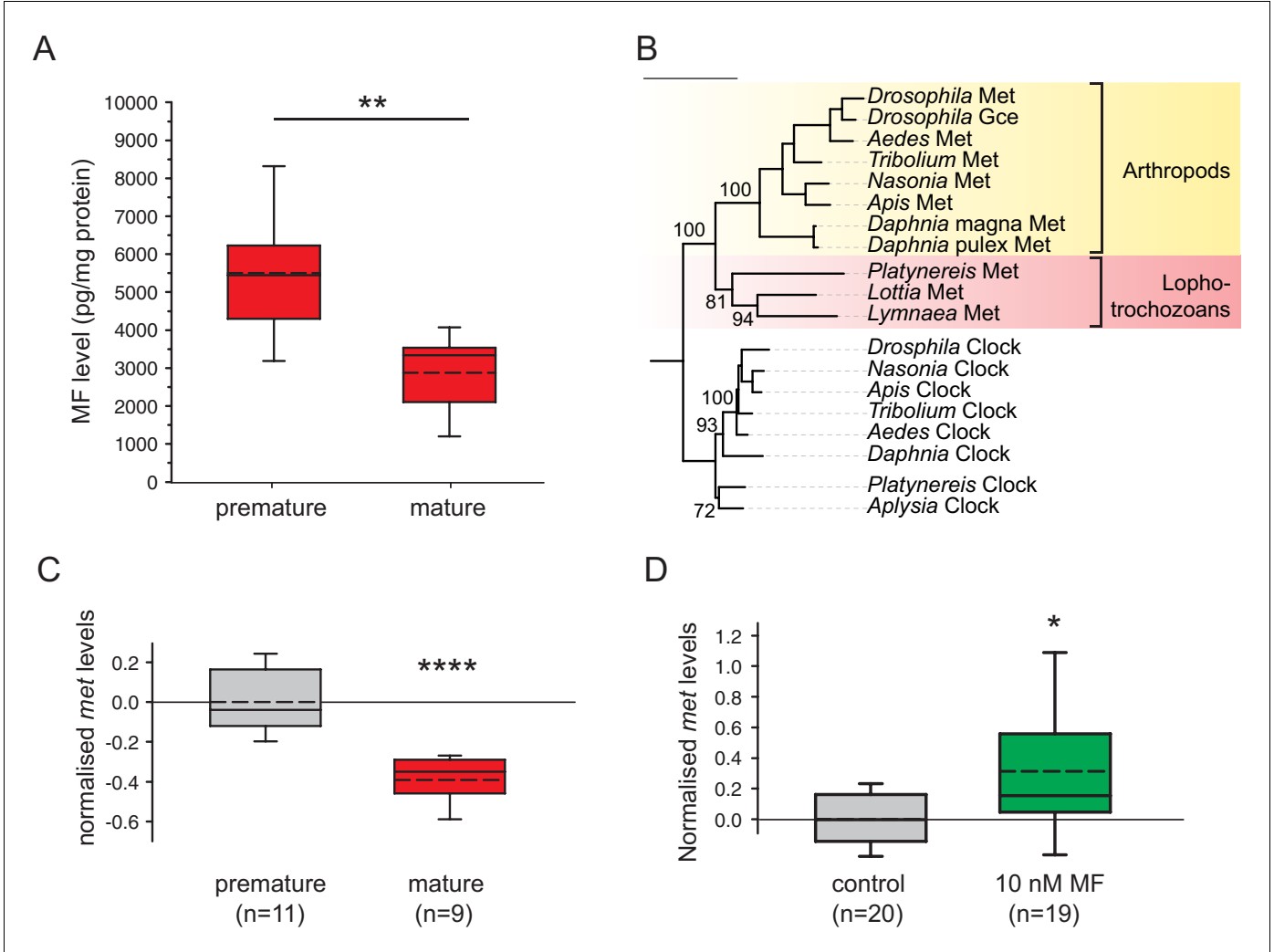

**Figure 3.** Levels of MF as well as the sesquiterpenoid receptor orthologue Met drop over the course of maturation. (A) Significant decrease of MF levels between premature and mature heads; boxplots show amounts of MF normalised to the total protein content of the respective head sample. Heads of premature animals contain ~5510 ± 660 (S.E.M) pg MF per mg protein, whereas heads from mature animals contain ~2870 ± 495 pg MF per mg protein, thus roughly 50% of the amount found in heads of premature animals. Amounts are corrected for a MF recovery rate of 85%. The boxes show the first and third quartile, the median (solid line), and the mean (dashed line). Whiskers denote the 10th and 90th percentile. Significance was tested with a one-sided t-test, **$p < 0.01$. n: number of biological replicates. (B–D) *Platynereis* possesses an orthologue of arthropod sesquiterpenoid receptors that shows co-regulation with MF titres. (B) Maximum Likelihood tree supporting the identity of *Platynereis* Met as orthologue of the validated arthropod sesquiterpenoid receptors Methoprene-tolerant (Met) and Germ cells expressed (Gce). Phylogeny reconstructed with IQ-TREE 1.3.12 (*Minh et al., 2013*; *Nguyen et al., 2015*). By parameter optimisation, the LG+F+I+G4 substitution model was selected. Numbers refer to confidence levels (in %) of major nodes derived from 1000 replicates. Accessions: *Drosophila* Met: NP_511126.2; *Drosophila* Gce: NP_511160.2; *Aedes* Met: AAW82472.1; *Nasonia* Met: XP_001606775.2; *Apis* Met: XP_395005.4; *Tribolium* Met: NP_001092812.1; *Daphnia pulex* Met: BAM83853.1; *Daphnia magna* Met: BAM83855.1; *Platynereis* Met: KU756288 (this study); *Lottia* Met: XP_009046608.1, *Lymnaea* Met: FX197139.1. *Drosophila* Clock: NP_523964.2; *Nasonia* Clock: XP_008214438.1; *Apis* Clock: XP_394233.1; *Aedes* Clock: XP_001662706.1; *Tribolium* Clock: NP_001106937.1; *Daphnia* Clock: EFX7997.1; *Platynereis* Clock: AGX93013.1; *Aplysia* Clock: XP_005112430.1. (C) Eleocyte *met* levels drop by ~39% in the course of maturation. qRT-PCR analysis of *met* transcript levels in eleocytes sampled from premature and mature animals. Relative *met* levels were calculated with respect to the arithmetical mean of the reference genes *sams* and *cim6pr,* and normalised to the level of the premature sample. (D) Eleocyte *met* levels are increased by the addition of MF. Similar analysis as in (C), indicating that *met* levels in MF-treated cultured eleocytes (right) are ~30% higher than in untreated controls (left). Relative *met* levels were calculated with respect to the reference gene *rps9*, and normalised to the level of the control sample. Boxplots (C,D) show the first and third quartile, the median (solid line), and the mean (dashed line). Whiskers denote the 10th and 90th percentile. Statistical significance in (B,C) was determined by a two-sided t-test.*: $p < 0.05$. ****$p < 0.0001$. n: number of biological replicates. Raw data for panels A, C,D provided in *Figure 3—source data 1*. Alignment for panel B provided as *Figure 3—source data 2*.

The following source data and figure supplements are available for figure 3:

*Figure 3 continued*

**Source data 1.** Data for the graphs in *Figure 3A* (MF content in pg normalised to mg protein per head), *Figure 3C* (*met* expression levels normalised to premature eleocytes) and *Figure 3D* (*met* expression levels normalised to the control treatment); data for the graphs in *Figure 3—figure supplement 2* and *Figure 3—figure supplement 3* (MF content in pg per head), *Figure 3—figure supplement 4A* and *Figure 3—figure supplement 4B*(*met* expression levels quantified against individual reference genes).
**Source data 2.** Alignment file for the phylogenetic tree of Met homologs (*Figure 3B*).
**Figure supplement 1.** Quantification of MF in the heads of *Platynereis* by GC-MS.
**Figure supplement 2.** The levels of MF per head drop over the course of maturation.
**Figure supplement 3.** Methylfarnesoate is found in the heads of earthworms.
**Figure supplement 4.** The choice of reference genes does not impact on the observed down-regulation of *met*.

this animal group (*Bellés et al., 2005*; *Jindra et al., 2015*). To determine whether the *Platynereis* MF system originated independently, or relies on common descent with arthropod sesquiterpenoid signalling systems, we investigated if *Platynereis* also possesses an orthologue of arthropod sesquiterpenoid receptors. Based on genetic screens in *Drosophila,* and subsequent biochemical and bioinformatics studies, homologues of the bHLH-PAS-domain-containing transcription factor Methoprene-tolerant (Met) have been identified as functional arthropod sesquiterpenoid receptors (*Miyakawa et al., 2013*; *Jindra et al., 2015*).

In transcriptome data available from *Platynereis* eleocytes, we identified a transcript encoding a full-length bHLH-PAS-domain-containing transcription factor that showed strong similarity to insect Methoprene-tolerant receptors. Molecular phylogeny revealed this protein to be a clear orthologue of the insect Methoprene-tolerant and germ cell expressed (Gce) receptors, as well as the functionally confirmed Met receptors from the crustacean *Daphnia*. We therefore named the corresponding *Platynereis* gene *methoprene-tolerant (met)*. The phylogenetic analysis also revealed Met orthologues in the molluscs *Lottia gigantea* and *Lymnaea stagnalis* (*Figure 3B*).

Using qRT-PCR analysis, we could confirm the expression of *Platynereis met* in eleocytes. Furthermore, these analyses revealed that the expression level of this gene dropped significantly during maturation, reaching only ~ 61% of the level observed in cells from premature animals (*Figure 3C* and *Figure 3—figure supplement 4*). This parallels the observed decline in MF concentration in the head measured by GC-MS (*Figure 3A*). Furthermore, treatment of cultured premature eleocytes with 10 nM MF induced the expression of Platynereis met, resulting in a 30% up-regulation (*Figure 3D*). Together, these data confirm the expression of met in eleocytes by an independent method and are also consistent with a positive autoregulation of this putative receptor by MF.

## Methylfarnesoate exerts a dominant repressive effect on vitellogenesis *in vivo*, and worms are susceptible to insecticides targeting the pathway

*In vivo*, eleocytes are freely floating in the coelomic cavity, compatible with the notion that they are accessible to hormonal factors like MF. However, coelomic fluid also represents a major metabolic compartment, where nutrients and metabolites are exchanged (*Fischer, 1979*; *Baert and Slomianny, 1987*; *Fischer and Hoeger, 1993*; *Hoeger and Kunz 1993*; *Hoeger et al., 1999*). This raises the possibility that other factors in the coelomic fluid could modulate or even block the effect of MF that we observe in isolated cell culture and prompted us to investigate the effect of MF on eleocytes in their natural context.

To reliably compare different animals, we used a classical decapitation paradigm. Decapitation leads to accelerated maturation, and, importantly, synchronisation of maturation across different individuals (*Hauenschild, 1966*, *1974*). We then quantified the maturation stage of MF-treated and DMSO-treated animals five days after decapitation, taking advantage of our established *vtg* qRT-PCR assay (*Figure 4A* and *Figure 4—figure supplement 1*).

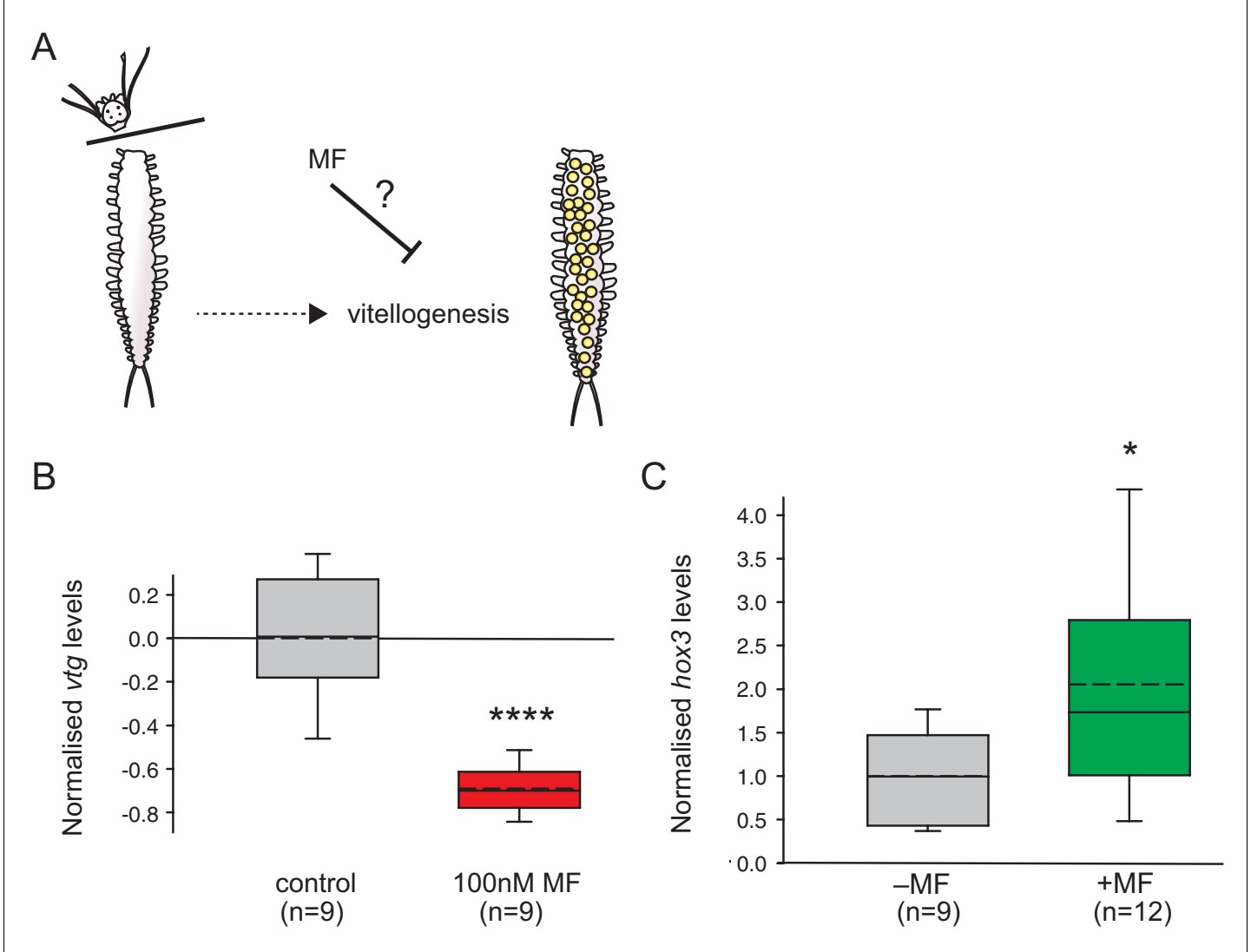

**Figure 4.** MF also represses *vtg* expression *in vivo*, and sustains expression of a marker for caudal growth upon posterior amputation. **(A)** Setup of experiment testing the ability of MF to interfere with vitellogenesis *in vivo*. Following decapitation, female individuals are known to start vitellogenesis, reflecting loss of nereidin; treatment with MF tests if the repressive effect of nereidin on *vtg* observed in primary cultures of coelomic cells also occurs in the normal context of these cells. **(B)** qRT-PCR analysis of *vtg* expression in decapitated animals 5 days after decapitation. The animals were either treated with 0.1% DMSO (control) or in 100 nM MF in 0.1% DMSO for five days. Expression levels of *vtg* upon presence of MF are ~70% lower than in treated animals. *vtg* levels were related to the arithmetical mean of the reference genes *rps9* and *sams* and normalised to the *vtg*-expression level of the control. **(C)** MF sustains expression of *hox3*, a marker diagnostic for caudal regeneration. qRT-PCR analysis of *hox3* expression in *Platynereis* fragments 5 days after posterior amputation. Headless worm fragments treated with 100 nM MF in 0.1% DMSO (right) show twice as high levels of *hox3* than headless control fragments (left) only treated with 0.1% DMSO. *Hox3*-expression is relative to that of the arithmetical mean of the reference genes *rps9* and *sams* and normalised to the *hox3*-expression level of the untreated samples. **(B,C)** Boxplots show the first and third quartile, the median (solid line), and the mean (dashed line). Whiskers denote the 10th and 90th percentile. Statistical significance was tested by a one-sided **(B)** and a two-sided t-test **(C)**, respectively. *p<0.05. ****p<0.0001. n: number of biological replicates. Raw data for panels B,C provided in *Figure 4—source data 1*.

The following source data and figure supplement are available for figure 4:

**Source data 1.** Data for the graphs in *Figure 4B* and *Figure 4C* (*vtg/hox3* expression levels normalised to the respective control treatment).

**Figure supplement 1.** The choice of reference genes does not impact on the observed down-regulation of *vtg*.

The results of these experiments showed that *vtg* levels were strongly down-regulated in MF-treated worms when compared to a non-treated control group (*Figure 4B*). This establishes that MF does not only have an effect on isolated coelomocytes, but also exerts a similar effect *in vivo*, in spite of other physiological regulators that might be present in the coelomic fluid. Notably, our data are also reminiscent of classical decerebration and re-transplantation experiments that showed that implanted brains could reverse the accelerated maturation observed after decerebration (*Hauenschild, 1956a*, *1956b*, *1966*; *Golding, 1983*). This implicates MF as a major hormonal cue responsible for the anti-maturation effect of the juvenile brain.

The sesquiterpenoid pathway is a prime target for a large variety of insecticides. After observing the effect of MF on annelid reproduction both *in vivo* and *in vitro*, we tested whether the juvenile hormone analogue methoprene is also capable of influencing vitellogenesis in *Platynereis*. We treated decapitated animals as before and assessed changes in *vtg* levels. In this experimental setup, we did not observe a significant decrease in *vtg* levels. In contrast, applying methoprene to eleocytes in our cell culture assay clearly led to reduced *vtg* levels at a concentration of 10 nM (*Figure 5*). To assess if this effect was specific to methoprene, we also tested the structurally unrelated, yet highly used insecticide pyriproxyfen for its ability to interfere with endogenous *vtg* transcript levels. Indeed, when applying pyriproxyfen at the same concentration (10 nM), we again observed a significant decrease in the abundance of *vtg* transcripts when compared to controls. This indicates a profound effect of presumed insect-specific growth regulators on a non-target species.

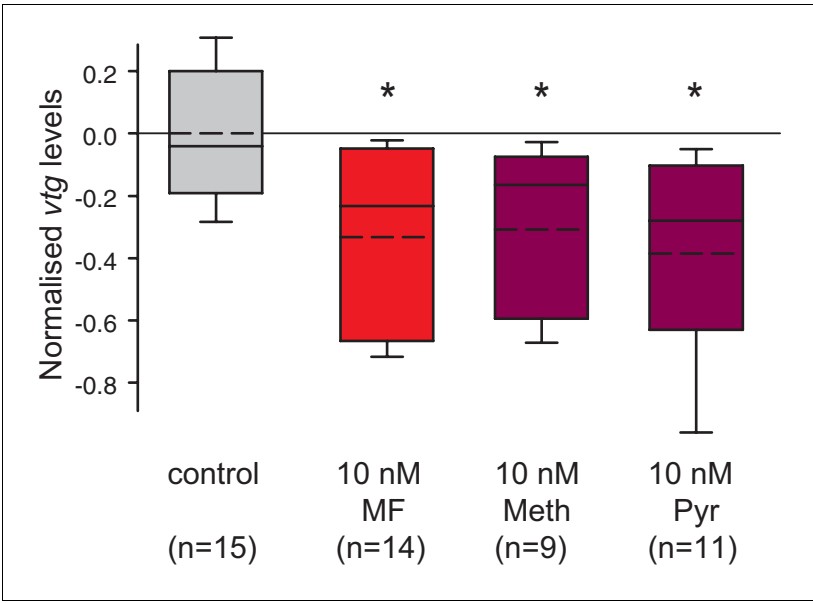

**Figure 5.** *Vitellogenin* expression in worm coelomocytes is suppressed by the hormone agonists methoprene and pyriproxyfen that are considered to act exclusively on insects. Expression levels of *vtg* in coelomocytes are significantly down-regulated by treatment with 10 nM methoprene (Meth) and 10 nM pyriproxyfen (Pyr) in 0.01% DMSO. The graph shows qRT-PCR quantification of vtg levels after over night (20 hr) treatment of primary coelomocyte cultures; expression levels are relative to those of the reference gene *rps9* and are normalised to the *vtg*-expression level of the control (DMSO-treated cells). The Boxplots show the first and third quartile, the median (solid line), and the mean (dashed line). Whiskers denote the 10th and 90th percentile. Statistical significance was tested first by an ANOVA, followed by a two-sided t-test with adjustment of the p-values for multiple testing to assess differences between the different groups. *$p < 0.05$. n: number of biological replicates. Raw data provided in *Figure 5—source data 1*.

The following source data is available for figure 5:

**Source data 1.** Data for the graph in *Figure 5* (*vtg* expression levels normalised to the control treatment).

## Methylfarnesoate sustains expression of a marker diagnostic for caudal re-growth

In addition to its central role in suppressing maturation, nereidin has also been classically defined by the ability of transplanted brain pieces to maintain caudal regenerative capacity (*Hauenschild, 1974*; *Hofmann, 1976*). Both functions have been used interchangeably in nereidin research. However, the lack of molecular candidates for nereidin has prevented the assessment of whether both functions rely on identical molecules, or if they can be separated. In a final set of experiments, we therefore tested if MF had any impact on regeneration of caudal body segments. To minimise inter-individual variations in regenerative capacities (*Golding, 1967b*), we followed a similar approach as *Hauenschild (1974)* in which hormone-exposed and control fragments were derived from the same animal. In addition, as in our maturation assays, we aimed to develop a quantitative approach to assess the molecular effect of MF on caudal regeneration. For this, we established a qRT-PCR assay quantifying the expression of the homeobox gene *hox3*. *Hox3* expression has earlier been demonstrated to be a highly specific early on marker in caudal regeneration (*Pfeifer et al., 2012*). More specifically, it has recently been linked to ectodermal cells of likely stem cell character (ectoteloblasts, *Gazave et al., 2013*) involved in the regenerative process. Our results show that the expression of *hox3* is significantly up-regulated (~2 fold) after 5 days in test fragments treated with 100 nM MF when compared to DMSO-treated control fragments (*Figure 4C*). This finding is consistent with the idea that besides its role in repressing maturation, MF may also contributes to sustaining putative stem cells involved in regeneration.

## Discussion

Together, our results establish methylfarnesoate as a critical neurohormone suppressing maturation in a key lophotrochozoan reference species. Our data are fully consistent with properties classically attributed to nereidin: in light of the massive amounts of Vitellogenin that are produced by animals committed to reproduction, the ability of MF to suppress *vtg* levels is compatible with a central role of this compound in controlling reproduction-associated energy expenditure, as expected for nereidin. Likewise, our quantifications of MF levels over the course of reproductive development (*Figure 3A*) are consistent with the decline in nereidin activity present in transplanted heads of different stages. Although our data clearly argue against the long-held assumption that nereidin is a neuropeptide, they are still consistent with the chemical properties (apparent mass, lipophilic nature) that have been derived from previous purification attempts. Likewise, the data are consistent with the notion that related hormones are present in other annelids (see below).

Our findings not only reveal an overall role of this sesquiterpenoid in governing energy expenditure but also highlight critical mechanistic aspects: (i) We show that eleocytes are direct target cells for MF, indicating a direct cross-talk between the brain and a highly relevant metabolic cell type. (ii) Moreover, the observed effect of MF on *vitellogenin* levels is consistent with a direct function of MF in controlling this key factor for oocyte maturation. (iii) The positive autoregulation of the sesquiterpenoid receptor orthologue Met that we observe is reminiscent of other hormone ligand-receptor systems, such as thyroxin or ecdysone (*Thummel, 1995*; *Bagamasbad and Denver, 2011*). In the biological context of semelparous reproduction, such autoregulation provides an attractive model how graded changes in MF levels can be amplified, contributing to sharpening the response of the target cells to above- vs. below-threshold levels.

Future work on the fine-tuning of MF regulation, and its effect on target cells, will likely yield more detailed insight into the role of MF for eleocyte physiology. Such analyses will extend our understanding of molecular changes in the eleocyte, for instance to cover genes involved in nucleotide metabolism (*Hoeger et al., 1999*). Moreover, they will resolve the question if residual levels of MF signalling are required for proper vitellogenesis, or if vitellogenesis only takes place in complete absence of MF. This addresses classical debates about the requirement of nereidin (*Hauenschild, 1966*; *Durchon and Porchet, 1970*; *Golding, 1983*) but is also interesting for cross-comparisons between the annelid system and insects, where sesquiterpenoids do not only define juvenile stages – similar to *Platynereis* – but in later life play a – positive – role for vitellogenesis itself.

A second phenomenon where the identification of MF provides important mechanistic insight, and opens up new experimental possibilities concerns the question of how regenerative abilities of *Platynereis* and other species are controlled. From a global perspective of organismal energy

expenditure, it is plausible that semelparous reproduction (investment in growth of germ cells) and regeneration (investment in re-growth of soma) are anti-correlated, although it remains unclear exactly how this is achieved at the molecular level. Our finding that MF sustains expression of a marker (*hox3*) characteristic for a population of putative stem cells in posterior segment addition (*Pfeifer et al., 2012*; *Gazave et al., 2013*) is consistent with the idea that MF also plays a role in regenerative control, but in contrast to *vtg* regulation in eleocytes, it is less clear if this effect is direct or caused by other mediating factors. Due to the complexity of regeneration, reliable and quantitative short-term assays for regeneration will need to be established to pursue this question.

The identification of MF in *Platynereis* heads challenges several hypotheses: most directly, it defeats the hypothesis that nereidin is a peptide, a notion that has been propagated in the literature for more than 30 years (*Cardon et al., 1981*; *Durchon, 1984*). Furthermore, MF and related sesqui-terpenoid hormones have so far only been found in arthropods (*Bellés et al., 2005*; *Jindra et al., 2015*), and their evolution has been hypothesised to be a secondary consequence of the loss of the sterol synthesis pathway in this clade (*Zandee, 1964*; *Bellés et al., 2005*). As marine nereidid worms were previously shown to possess a functional sterol synthesis pathway (*Wootton and Wright, 1962*), and *Platynereis* even has a functional estrogen receptor (*Keay and Thornton, 2009*), our dis-covery makes it unlikely that loss of sterol synthesis preceded the evolution of sesquiterpenoid hor-mones. Rather, it suggests that, like sterols, bioactive sesquiterpenoids, could have been part of the ancient molecular repertoire available in the common ancestor of protostomes, if not even at earlier stages of animal evolution.

Several lines of experimental evidence support this notion: Exogenous MF was found to affect lar-val development of the annelid *Capitella sp.* (*Laufer and Biggers, 2001*), and annelid extracts dis-played biological activity in assays for insect juvenile hormones (*Schneiderman and Gilbert, 1958*). The latter result would suggest that insect juvenile hormones might also be active in the annelid sys-tem, but this issue has not yet been addressed experimentally. Moreover, we have detected MF in head extracts of common earthworms (*Lumbricus sp.*, *Figure 3—figure supplement 3*). Beyond the annelid clade, our identification of Methoprene-tolerant orthologues in the limpet *Lottia* and the pond snail *Lymnaea* indicates that the sesquiterpenoid signalling pathway might even exist in mol-luscs. As annelids and molluscs are large groups within the lophotrochozoan superphylum (*Halanych et al., 1995*), this implies a potential physiological role for sesquiterpenoids in a broad set of (mostly aquatic) species.

What could such physiological roles be? Semelparity, as it occurs in *Platynereis*, is considered an extreme form of 'capital breeding', a reproductive strategy where females build up large energy stores in their bodies that are then invested into offspring at particular time points (*Drent and Daan, 1980*). Capital breeding strategies allow animals to time their reproduction more precisely and have larger clutches of offspring than could be afforded by daily nutrient uptake. It likely represents an ancient reproductive strategy that is very common in poikilothermic animals (*Bonnet et al., 1998*). Therefore, it is plausible that the role for MF uncovered in the marine annelid *Platynereis* could reflect a more general role of sesquiterpenoids in regulating reproductive energy expenditure. The inhibitory function for maturation we uncover in *Platynereis* is reminiscent of the juvenoid functions of insect juvenile hormones: During pre-adult development, these provide a molecular cue for juve-nile status, preventing insects from entering into adulthood, and thereby indirectly preventing repro-ductive energy expenditure as well. Likewise, MF treatment of larval stages of crustaceans can delay metamorphosis or prevent gonad formation (*Borst et al., 1987*; *Tsukimura et al., 2006*).

During adulthood, however, insect and crustacean sesquiterpenoids have distinct functions. In contrast to our results in worms, juvenile hormones commonly act as positive regulators of vitello-genesis in the adult insect fat body, whereas their down-regulation is associated with diapause and longevity (*Gäde et al., 1997*; *Wyatt, 1997*; *Hansen et al., 2014*). Similarly, MF promotes adult mat-uration, vitellogenesis and mating behaviour in crustaceans (*Laufer et al., 1987*; *Homola and Chang, 1997*; *Subramoniam, 2011*), with a separate function in sex determination (*Toyota et al., 2015*). As *Platynereis* retains certain MF levels even in the mature stage (see above), it will be a suit-able model to address additional roles of MF in the life history of an annelid.

Finally, our discovery of a sesquiterpenoid hormone in an annelid is not only relevant to under-stand hormone evolution but it likely also has significant ecological implications. Reproductive hor-mones are attractive targets for pesticides. Juvenile hormone analogues like methoprene and later pyriproxyfen have initially been designed to interfere with insect development and are broadly used

to fight insect pests such as *Aedes*, a vector for dengue, chikungunya, Zika, and yellow fever viruses (reviewed in *Henrick, 2007*). The discovery the sesquiterpenoid MF in crustaceans by *Laufer et al. (1987)* already raised concerns if the effect of juvenile hormone analogues is restricted to insect development (*Walker et al., 2005*; *Abe et al., 2015*), calling for more specific tests with non-target species. Our results argue that aquatic lophotrochozoans should also be systematically included in specificity tests for compounds targeting sesquiterpenoid signalling pathway, and that such tests should not be restricted to mortality (*Henrick, 2007*), but include assays on reproductive fitness. For instance, in our studies, pyriproxyfen impacts on *vtg* levels at concentrations of 10 nM which lies in the range of pyriproxyfen found in surface waters (*Sullivan and Goh, 2008*). In turn, sesquiterpenoid analogues may also be suitable leads in the search for specific compounds targeting pest species outside the arthropod clade or factors to favourably affect the balance between reproductive and somatic growth in aqua- and mariculture.

Taken together, our data argue for a critical role of sesquiterpenoid hormones outside arthropods and indicate that methylfarnesoate is the major constituent of the long-sought brain hormone nereidin, a key factor in regulating maturation in marine capital breeders. The discovery of this hormone in an annelid provides a new entry point to understand hormonal regulation of energy expenditure and sheds new light on the evolution of animal hormone systems.

## Materials and methods

### Animals
*Platynereis dumerilii* were kept in a continuous culture at the Marine Facility of the Max F. Perutz Laboratories in Vienna (for details see *Hauenschild and Fischer, 1969*).

### Cell isolation
After anaesthetisation of the animals in a 1:1 (v/v) mixture of seawater and 7.5% (w/v) $MgCl_2$ in distilled water, the animals were transferred into 500 µL Nereis balanced salt solution (NBSS, see *Fischer et al., 1991*) and the coelomic cavity was opened with fine scissors. The coelomic cells were gently squeezed out, passed through 136, 70 and 30 µm gauze filters to select for eleocytes (>90% pure as judged by microscopical inspection). Cells were then collected by centrifugation (5 min, 500 ×g, 0°C), and washed once with NBSS. For the isolation of coelomic cells (eleocytes and oocytes), only the first gauze filtration was carried out.

### Bioassay
Coelomic cells were isolated as described above and cultured a total of volume of 50 µL of culture medium without hen egg ultra filtrate (*Schenk and Hoeger, 2010*) overnight at 19°C on a rocking platform, in a BSA-coated 96-well microtitre plate. All animals were in the premature stage, with oocyte diameters between 50 and 65 µm, i.e. before the irreversible onset of maturation assumed to be beginning from oocyte diameters of 85 µm onwards (*Hauenschild, 1966*). Test substances were diluted in either 100 mM $Na_2PO_4$, pH 7.5, or in NBSS, and their final volume never exceeded 10% of the total, DMSO concentration was kept to a maximum of 0.01%. Controls received only the vehicle.

After incubation, cells were harvested by centrifugation (5 min, 500 xg, 0°C) and total RNA was isolated with a commercial kit (RNeasy-Kit, Qiagen, Hilden, Germany) according to the manufacturer's protocol, stored at −80°C and used for qRT-PCR using primers against *Platynereis vitellogenin* (sequences see *Table 1*) to assess changes in *vitellogenin* transcript abundance. As *vitellogenin* expression is strongly dependent on the maturation stage (see results) and as individual animals can show considerable variations in their physiology (*Fischer and Hoeger, 1993*; *Hoeger and Kunz 1993*) the cells of at least two animals were pooled per treatment group. Each assay consisted of three biological replicates and was at least repeated once.

### qRT-PCR analyses
Primers for qPCR were designed using the 'Universal Probe Library' web page provided by Hoffman-LaRoche (http://lifescience.roche.com, see *Table 1*). Available genome information (http://4dx.embl.de/platy/) was used to choose primers across exon/intron junctions. Prior to use, all primers were tested for their dynamic range by serial dilutions (10x, 100x, 1000x) of cDNA. These tests also

**Table 1.** List of primers used for qRT-PCR, including predicted melting temperature.

| Name | Sequence 5'→3' | $T_m$ / °C | Efficiency |
|---|---|---|---|
| Pdu vtg qPCR1 F | ACAGGCCATCACATTCACAA | 56.4 | 101% |
| Pdu vtg qPCR1 R | TCTGCTCACGTCTCTTTCCA | 58.4 | |
| Pdu met qPCR1 F | GGATGATTATGATGTATACCTGCAAC | 62.9 | 102% |
| Pdu met qPCR1 R | AGACCGAACTGGCGTTTG | 56.3 | |
| Pdu hox3 qPCR1 F | CTACCCCTGGATGAGGGAAT | 60.5 | 95% |
| Pdu hox3 qPCR1 R | ACTTCCGGTTCCTGGTCC | 58.4 | |
| Pdu rps9 F | CGCCAGAGAGTTGCTGACT | 59.5 | 102% |
| Pdu rps9 R | ACTCCAATACGGACCAGACG | 60.5 | |
| Pdu sams qPCR1 F | CAGCAACGGTGAAATAACCA | 56.4 | 101% |
| Pdu sams qPCR1 R | CATCACTCACTTGATCGCAAA | 57.5 | |
| Pdu cim6pr qPCR1 F | ACTTCCCCTGCTGATGAGTG | 60.5 | 99% |
| Pdu cim6pr qPCR1 R | TTCGTAAGTCAGGTTTCCATTG | 58.4 | |
| Pdu cdc5 F | CCTATTGACATGGACGAAGATG | 60.1 | 100% |
| Pdu cdc5 R | TTCCCTGTGTGTTCGCAAG | 57.5 | |

included included a –RT-control (no reverse transcription). All selected primers showed a linear amplification profile in the tested dilution range, and the primer efficiencies were calculated ($E = 10^{1/slope} \times 100$, where "E" is the primer efficiency and "slope" is the calculated slope of the dilution series) to be between 95% and 102% (*Table 1*), and amplification from –RT controls was not observed. In these primer tests, amplified products were subcloned and the sequence confirmed by Sanger sequencing as described below.

In all assays, the same RNA amount was used for each group to be compared. Generally, this was 5 ng for the assessment of *vtg*, 150 ng for *met*, and 300 ng for *hox3*.

cDNA for qRT-PCR was synthesised using Qiagen's QuantiTect-Kit and final dilution of the cDNA to 50 μL. Briefly, the desired amount of RNA was diluted to 12 μL with nuclease-free $H_2O$, and 2 μL of the Kit's gDNA-Whipeout buffer (DNase I) were added. The DNase digest was performed for 6 min at 42°C, following cooling to 0°C, cDNA was then synthesised in a total volume of 20 μL for 15 min at 42°C, the reaction was stopped by heat deactivation of the enzymes (5 min, 95°C); finally, the reaction was centrifuged (1 min, 17.000 ×g), diluted to 50 μL with $H_2O$ and used for qRT-PCR.

qRT-PCR was performed on a Applied Biosciences StepOne instrument using SybrGreen (Thermo Fisher Scientific, Vienna, Austria) as reporter dye. Five microlitres cDNA were measured in a volume of 20 μL, with two technical replicates per sample. After an initial denaturation for 10 min at 95°C, amplification and reading of the reporter dye were carried out at 60°C for 1 min over a total of 40 cycles with a 3 s denaturation step at 95°C at the beginning of each cycle, this was followed by recording a melting curve of the amplified product (60.0–95.0°C in 0.3°C increments) to ascertain only a single amplified product.

Suitable reference genes were chosen based on (i) the stability of their expression values in the respective tissue and (ii) similar $C_t$-values for these genes in relation to the target gene. Specifically, the target and the reference gene were aimed to be within a $\triangle C_t$ -range of ±5. Generally, two reference genes were used for normalisation, and the target gene was normalised by the $2^{-\triangle\triangle Ct}$ – method to the arithmetic mean of these two reference genes. For the coelomocyte cultures, as they were extensively used in the context of the brain hormone fractionation, the limited material available for each assay only allowed us to use a single reference gene, *rps9* (*ribosomal protein small subunit 9*). However, we tested the stability of *rps9* in coelomocytes against the reference gene *cdc5* (*cell-cycle-dependent serine/threonine protein kinase*) when we first assessed *vtg*-transcript levels in eleocytes of different maturation stages (*Figure 1C*). Separate quantification against either reference gene produced similar results (*Figure 1—figure supplement 2*). This is in line with the fact that both of these genes were previously shown to be suitable reference genes (*Dray et al., 2010*;

*Zantke et al., 2013*). Additionally, we also assessed the stability of *rps9* in cell culture by comparing the normalised $C_t$-values within each experimental series. These analyses demonstrated that there is a high stability of these values (SD of 3%) (*Figure 2—figure supplement 2*). For *Platynereis S-adenosylmethionine synthetase (sams*; accession: KX907200) and *cation-independent-mannose-6-phosphate-receptor (cim6pr*; accession: KX907201), independent calculations with the respective second housekeeping gene support their suitability as additional reference genes (*Figure 3—figure supplement 4*, *Figure 4—figure supplement 1*).

Data were excluded from the analysis if the $\triangle C_t$-values were ±1.0x of the mean of the respective treatment group, or if the melt curve analysis indicated ambiguous $T_m$ for the PCR products. In any case, at least five biological replicates were used for analysis.

## Brain hormone purification

*Platynereis dumerilii* were anaesthetised as described above and decapitated in front of the pharynx. The heads were suspended in 80% methanol containing 0.1% formic acid, and disrupted by three ultrasonic bursts for 30 s on ice and extracted twice for 15 min on ice. Insoluble material was pelleted for 15 min and 20,000 ×g at 0°C, after which the pellet was again extracted and centrifuged. To further eliminate insoluble macromolecules, the raw extract was left to precipitate for 30 min at −20°C, and was again centrifuged for 15 min and 20,000 ×g at 0°C. The resulting supernatant was mildly dried down *in vacuo* and the pellet suspended in 0.1% formic acid, by vigorous vortexing, followed by shaking at 1400 rpm at room temperature on a Thermomixer, and finally centrifuged (2 min, 17,000 ×g). The extract was then applied onto phenyl-extraction columns (Alltech, 100 mg sorbent, ordered from Thermo Fisher Scientific, Vienna, Austria) equilibrated in 0.1% formic acid. The flow-through was discarded, the column washed with 750 µL 0.1% formic acid, and eluted with 500 µL 50% methanol, 0.1% formic acid. The final eluate was dried down *in vacuo* and finally reconstituted at a concentration of 0.66 heads/µL in 100 mM $Na_2HPO_4$, pH 7.4, 10 min by shaking at 1400 rpm in a Thermomixer, and centrifuged (2 min 17,000 ×g). 6.0 head equivalents were used per well in the cell culture assay.

## Proteinase treatment

For proteinase treatment, an aliquot of the eluted active fraction was incubated with a final concentration of 1 mg/mL proteinase K (EC 3.4.21.64; Merck Millipore, Vienna, Austria; 30 mU) for 60 min at 56°C. The reaction was stopped by heating the sample to 95°C for 5 min and, after cooling to 0°C by making the sample 0.1 mM in phenylmethylsulphonyl fluoride, finally the samples were clarified by centrifugation. Controls without proteinase K were treated alike. Again, 6.0 head equivalents were used per cell culture assay.

## Brain hormone identification

The brain hormone enriched fraction was dried down *in vacuo*, suspended in 0.1% trifluoroacetic acid and resolved on a Symmetry-ODS-column (Waters) equilibrated in 20% solvent B (0.1% TFA in acetonitrile) and 80% solvent A (0.1% TFA in water), after an isocratic wash for 1.5 min, the column was developed by raising the concentration of B to 100% over 35 min, followed by a plateau of 3.5 min, with a flow rate of 0.5 mL/min. The eluent was monitored at 215 nm and 280 nm and a total of 16 fractions eluting were collected. The fractions were dried down *in vacuo* and stored at −20°C until use. To determine the biological activity of the collected peaks, each was reconstituted in 100 mM $Na_2HPO_4$, pH 7.5 at a concentration of 0.66 heads per µL and used in the cell culture assay as described above. Peaks showing a significant reduction in *vitellogenin* transcript abundance in two independent experiments were further analysed by LC-MS/MS, and the obtained elution profiles and fragment patterns were compared with authentic standards.

LC-MS/MS was performed on a LTQ Orbitrap (Thermo Fisher Scientific) equipped with a nano-spray ion source connected to a Dionex Ultimate 3000 UHPLC (Thermo Fisher Scientific), equipped with a Pepmap100 C18 guard column and a Pepmap100 C18 column (2 cm × 75 µm and 50 cm × 75 µm, respectively, 5 µm particle size, Thermo Fisher Scientific) equilibrated in 1% B (80% acetonitrile, 0.2% formic acid) and 99% A (0.2% formic acid in water). The samples were loaded onto the guard column with a flow rate of 10 µL/min and washed for 10 min using 100% A. The sample was the resolved on the analytical column with a flow of 300 nL/min. The column was developed for 10

min isocratic with 1% B; B was then raised to 20% in one min, before reaching 80% after 49 min. Following a 3-min plateau, the column was washed by raising B to 90% B in 2 min and washed for another 2 min. Mass spectra acquisition started after 10 min; precursor ions were scanned in the range of m/z 100–1000, and the 12 most abundant ions were fragmented in the HCD cell with a normalised fragmentation energy of 40% ± 75%.

## Quantification of brain hormone

Heads were collected as described above and stored at −80°C. They were thawed at room temperature, and subjected to 4 freeze-thaw cycle between liquid $N_2$ and 55°C warm water. The heads were then transferred into 1.5 mL auto sampler vials containing a glass bead and covered with 400 μL 80% methanol, and the head capsule was broken with four cycles in an ultra sonic water bath at 55°C for ten minutes. The homogenate was then extracted with 1000 μL heptane for 5 min with vigorous shaking at 1400 rpm at room temperature in a Thermomixer followed by centrifugation at 2500 ×g for 10 min. The organic phase was recovered and the aqueous phase re-extracted with 500 μL as above. The pooled extract was mildly dried down *in vacuo*, dissolved in 300 μL heptane, transferred into a 300 μL auto sampler vial insert, dried down, and finally dissolved in 10 μL heptane. The remaining aqueous phase was dried down as well, and used for protein determination (see below).

Brain hormone amounts were quantified by GC-MS on an Agilent 6890N GC equipped with an Agilent 5973 mass spectrometer. The GC was operated in splitless mode with constant pressure of 68.6 kPa and an initial flow of 1.0 mL/min with He-gas as carrier. After injection (inlet temperature: 250°C, purge flow: 50 mL/min, purge time: 2.0 min) and an initial plateau phase of 1 min at 90°C, the temperature was raised from 90°C for 280°C at 20 °C/min followed by a plateau for 7.5 min.

The eluting samples were ionised and fragmented by electron impact at an ionisation energy of 2376 eV and monitored in the range from m/z 50.0–400.0 with a sampling rate of $2^3$/s in the total ion channel, and for identification and quantification in selected ion mode (SIM) at m/z 114.0, 136.0 and 219.0 with 3.1 cycles/s and a dwell time of 100 ms per ion. The MS-source was operated at 230°C and the quadrupole at 150°C. One μL of the samples was injected, and the sample was resolved on a DB-5MS column (J&W Scientific, Agilent; 30 m × 0.25 mm, 0.25 μm film thickness).

The amount of methylfarnesoate was determined by comparison to authentic standards of methylfarnesoate (Echelon Biosciences). For quantification, serial dilutions of a 2.5 ng/μL sock solution of MF in heptane were measured. Each data point was measured in triplicate, and a linear regression was fitted. Peak detection and quantification was carried out with the program, MSD ChemStation (Agilent); peaks had to fit into a specified retention time window (± 0.1 min of that of the standard) and meet the qualifier ions. In rare cases, where automatic peak detection failed, peaks were manually integrated and the amount of MF calculated based on the regression generated.

The extraction efficiency of our method was assessed by using the heads of 25 young animals as a matrix and spiking in a total of 12.5 ng MF (out of a 12.5 ng/μL stock solution). The samples were then processed as described, and finally reconstituted in 25 μL heptane, out of which one μl was measured, resulting in an expected MF amount of 500 pg. The stock solution served as reference and was diluted 25-fold immediately before measurement. The extraction efficiency was thus estimated to be >85%. Under our conditions, the limit of detection was 10 fmole MF (2.5 pg MF/μL), and the limit of quantification ~ 60 fmole MF (~15 pg MF/μL).

## Earthworm head sampling

Earthworms *Lumbricus spec.* (Giant Canadian Night Crawlers, Baitmaster) were purchased at a local fishing bait shop (Angelsport Gangl, Wien). The animals were treated with antibiotics (125 mg/L ampicillin, 500 mg/L streptomycin sulphate) for 5 min, before the antibiotics solution was allowed to soak into the soil the earthworms were covered in. After 24 hr in the antibiotics treated soil, the animals were anaesthetised in 15% ethanol at 4°C for 15 min. Two to six heads (prostomium plus the next 4 segments) were collected on ice, snap frozen in liquid $N_2$, and stored at −80°C until further use. For the extraction of MF, the samples were subjected to four freeze-thaw cycles between 55°C and liquid $N_2$. Thereafter, they were covered in 80% MeOH and processed as described for *Platynereis* heads, with adjustments accounting for the larger tissue size. The finally obtained dried extract was reconstituted in n-heptane to yield ~2 heads per μL. GC-MS measurements were carried out as described for *Platynereis* heads.

### *In vivo* treatment of worms

For *in vivo* treatment of worms, premature female animals (max. Oocyte diameter < 65 µm) were selected and kept for 2 days in sterile filtered natural sea water (NSW). They were then decapitated (as described above) to synchronise and to set in motion sexual maturation (*Hauenschild, 1966*). The animals were transferred in groups of 4–5 into glazed ceramic bowls (treated with 6% PEG 6000 to prevent unspecific hydrophobic binding of the hormones to the glass coating) containing 50 mL sterile filtered NSW supplemented with 0.125 mg/mL ampicillin and 0.500 mg/mL streptomycin sulphate. Worms were treated either with 100 nM methylfarnesoate, 100 nM methoprene (Sigma, both added from a 100 µM stock in DMSO, resulting 0.1% DMSO final concentration), or 0.1% DMSO alone. After 24 hr, water was exchanged, and the antibiotics concentration reduced to 50%, after that, the water exchanged every 24 hr. After 5 days, the animals were snap-frozen in liquid $N_2$ without any residual water and stored at −80°C until RNA extraction.

### Protein determination

Protein concentration was determined by the bicinchoninic acid assay with BSA as standard (*Smith et al., 1985*).

### Regeneration assay

Fed, young premature (35–50 parapodial segments, pps) worms were kept in sterile-filtered NSW supplemented with 0.125 mg/mL ampicillin and 0.500 mg/mL streptomycin for 24 hr, and further for an additional 24 hr in half concentrated ampicillin/streptomycin. Generally, the regeneration assay was carried out as described by *Hauenschild (1974)*. Briefly, worms were anesthetised as described and cut three times between segments. The first cut was after the 21st pps, the second after the 26th pps, and the third cut was after the 31st pps producing fragments of five segments. The test fragments from each individual animal used for assays consisted of pps 22–26, and 27–31. The fragments were then kept in 1 mL sterile-filtered NSW containing half concentrated ampicillin/streptomycin in GC-auto sampler vials containing either 100 nM methylfarnesoate (added from a 100 µM stock solution in DMSO, fragment '+MF'), or vehicle alone (0.1% DMSO, fragment '−MF'); during the course of the experiment, the water was exchanged every 24 hr, care was taken to not let the worm fragments dry out. To rule out any effect of the position within the animal on regenerative abilities, +MF and control were alternated between the anterior and posterior fragments. The anterior and posterior +MF- and control fragments of two individual worms were combined, to serve as one biological replicate for +MF, and −MF treatment groups, respectively. They were snap-frozen in liquid nitrogen, stored at −80°C and used for the quantification of hox3 via qRT-PCR.

### Gene identification and confirmation

A transcript sequence encoding *Platynereis* Vitellogenin (accession no. KU756287) was identified from an eleocyte-specific transcriptome dataset (Schenk et al., unpublished), and sequence-validated after PCR cloning using the primers 5'-ATGAAGACTCTCCTGATCTTCG-3' and 5'-CTAGTAGTAGAATCTTGGTCCTTCAC-3' (for the list of used sequencing primers see *Table 2*).

Similarly, we identified a transcript encoding *Platynereis* Methoprene-tolerant (accession no. KU756288) that was confirmed by sequencing of a fragment sub-cloned with the primers 5'-ATGGAGCCGAATTCGGAGCAGAATTCGG-3' and 5'-TCAACATGTCTCAGTTTCTTTTTGAGCG-3'.

For cloning, cDNA was prepared with the Transcriptor High Fidelity cDNA Synthesis Kit (Roche) according to the manufacturer's instructions with oligo dT-priming from eleocyte or whole worm RNA (isolated with the RNeasy Kit, Qiagen).

PCR was performed using Phusion Taq (Thermo), using the following cycle: 1x {98°C, 30"}; 35x {98°C, 7"; x°C, 20"; 72°C, 30"/kb}; 1x {72°C, 7'30"}; 10°C. The annealing temperature 'x' was determined by the web page 'OligoCalc' (http://www.basic.northwestern.edu/biotools/oligocalc.html), using on the 'salt adjusted' algorithm. If necessary, annealing temperatures were optimized by gradient PCR and adjusted accordingly.

The PCR products were run on agarose gels, stained with SybrSafe (Invitrogen) DNA-dye, the specific products were cut out and purified using the QIAquick gel extraction kit (Qiagen). The purified PCR-products were sub-cloned into a pJET-vector (CloneJet Kit, Thermo), and transformed into NEB-5-alpha (New England Biolabs) chemical competent *E. coli*. Bacteria plated on Luria-Bertani

**Table 2.** List of primers used for cloning and sequence validation.

| Name | Sequence 5′→3′ | $T_m$ / °c |
|---|---|---|
| Pdu vtg 1F | ATGAAGACTCTCCTGATCTTCG | 60.1 |
| Pdu vtg 1R | CTAGTAGTAGAATCTTGGTCCTTCAC | 64.6 |
| Pdu vtg_seq 1F | AGCCCTAGAAGCTGCCTCTG | 62.5 |
| Pdu vtg_seq 2F | ATTGCTCAATCTGAACTCCCATGC | 63.6 |
| Pdu vtg_seq 3F | GCTGTTCCACAGGAAATTGC | 58.4 |
| Pdu vtg_seq 4F | GCTTTGGTCAGTGGACTTCC | 60.5 |
| Pdu vtg_seq 1R | GGCAATCCTCTGATGTAAACATTCTC | 64.6 |
| Pdu vtg_seq 2R | CAAGCGTTTCACGACCAAGAGG | 64.2 |
| Pdu vtg_seq 3R | GAAGAGCTTCTTGCTGGAGC | 60.5 |
| Pdu vtg_seq 4R | AAGACCAGCTGGCGCGTTATG | 63.2 |
| Pdu met F | ATGGAGCCGAATTCGGAGCAGAATTCGG | 71.8 |
| Pdu met R | TCAACATGTCTCAGTTTCTTTTTGAGCG | 65.6 |
| Pdu hox3 F | CCCCGGGGGCTCTTGGTTTT | 61.6 |
| Pdu hox3 R | GCCATCTCTATTCTCCTCGGCCG | 68.3 |

(LB)-medium agar plates supplemented with ampicillin (amp, 50 µg/mL), after growth over night single colonies were selected and grown over night in LB[Amp]-medium. The next day, plasmids were purified (*Birnboim and Doly, 1979*) and submitted for Sanger sequencing using the indicated primers.

## Gene orthology and domain analyses

For domain analyses of *Platynereis* Vtg and lipoproteins representatives for other clades, we used the domain annotation implemented in SMART (http://smart.embl.de; *Schultz et al., 1998*; *Letunic et al., 2015*). For determining phylogenetic relationships, we aligned full-length protein sequences representative of distinct clades using MAFFT v7.221 (*Katoh and Standley, 2013*), and constructed Maximum Likelihood trees with IQ-TREE 1.3.12 (*Minh et al., 2013*; *Nguyen et al., 2015*). The consensus tree was visualized using iTOL (http://itol.embl.de/, *Letunic and Bork, 2011*).

For assessing the phylogenetic relationships of *Platynereis* Met and other bHLH-PAS domain proteins, BLASTP searches were performed against the nr section of the NCBI sequence repository. Sequences from representatives of selected clades were aligned using MAFFT v7.221 (*Katoh and Standley, 2013*), and maximum likelihood phylogenetic trees were constructed with IQ-TREE, using the built-in parameter optimisation (*Minh et al., 2013*; *Nguyen et al., 2015*).

## Data analysis and statistics

All statistical tests were performed with the program R (*R Core Team, 2015*). First, all data were tested for normal distribution using the Shapiro-Wilk normality test. Statistical significance was then tested with a t-test incorporating Welch's correction to account for heteroscedasticity. If more than one comparison was to be assessed, statistical significance was first tested by a one-way analysis of variance test (ANOVA), and if significance was given, pair wise comparisons were carried out by a (Welch-corrected) t-test and the obtained p-values were adjusted for multiple testing by the method of Benjamini and Hochberg implemented in R yielding adjusted p-values ($p_{adj}$). A result was considered as statistically significant if $p<0.05$ in the case of t-tests, and $p<0.10$ in the case of ANOVA, the following t-test were then again considered to yield a statistically significant result if $p_{adj} < 0.05$. Significance levels of p-values and adjusted p-values are indicated by *$p<0.05$, **$p<0.01$, ***$p<0.001$, ****$p<0.0001$, if not stated otherwise.

## Acknowledgements

The research leading to these results has received funding from the European Research Council under the European Community's Seventh Framework Programme (FP7/2007–2013)/ERC Grant Agreement 260304 (to FR). FR and CG acknowledge support by the interdisciplinary Research Platform 'Marine Rhythms of Life' of the University of Vienna; SS was supported by a Lise Meitner-fellowship of the FWF (M1478-B19). The authors are indebted to Agne Valinciute for excellent technical help and for cloning and sequence-validation of the *met* gene, the Marine Facility staff at MFPL for animal husbandry, all members of the Raible and Tessmar labs for critical input, Dr. Dietrich K. Hofmann (Ruhr-Universität Bochum) for supply with brain hormone literature and continuous discussion, Dr. Ulrich Hoeger (Universität Mainz) for access to HPLC equipment and support, and Dr. Kristin Tessmar-Raible, Dr. Stephanie Bannister, and Dr. Graham Warren (all at the MFPL, Vienna) for helpful comments on the manuscript.

## Additional information

### Funding

| Funder | Grant reference number | Author |
| --- | --- | --- |
| Austrian Science Fund (FWF) | M1478-B19 | Sven Schenk |
| Universität Wien | Research Platform Marine Rhythms of Life | Christopher Gerner Florian Raible |
| European Research Council | Seventh Framework Programme (FP7/2007-2013)/ERC Grant Agreement 260304 | Florian Raible |

The funders had no role in study design, data collection and interpretation, or the decision to submit the work for publication.

### Author contributions

SS, FR, Conception and design, Acquisition of data, Analysis and interpretation of data, Drafting or revising the article; CK, Acquisition of data, Drafting or revising the article; PF, Acquisition of data, Analysis and interpretation of data, Drafting or revising the article; CG, Conception and design, Analysis and interpretation of data, Drafting or revising the article

### Author ORCIDs

Sven Schenk, http://orcid.org/0000-0002-7689-5854
Florian Raible, http://orcid.org/0000-0002-4515-6485

### Ethics

Animal experimentation: This study only used invertebrate worms from a laboratory culture. These are not covered by the specific ethical regulations applicable to higher invertebrates or vertebrates under Austrian and European legislation.

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
