## [Decision Letter]

[Editors’ note: this article was originally rejected after discussions between the reviewers, but the authors were invited to resubmit after an appeal against the decision.]

Thank you for submitting your work entitled "Discovery of methylfarnesoate as the annelid brain hormone reveals an ancient role of sesquiterpenoids in reproduction" for consideration by *eLife*. Your article has been favorably evaluated by Diethard Tautz (Senior Editor) and two reviewers, one of whom, Paul G Falkowski (Reviewer #1), is a member of our Board of Reviewing Editors, and another is Elisabeth Marchal (Reviewer #2).

Our decision has been reached after consultation between the reviewers. Based on these discussions and the individual reviews below, we regret to inform you that your work will not be considered further for publication in *eLife*.

The two reviewers had seemingly different specific concerns, but both clearly had reservations about the impact of the paper. The first reviewer was concerned that the identification of a signaling molecule, while interesting, is not sufficient for publication in *eLife* without some clear mechanistic understanding of how the molecular basis works. The second reviewer was concerned about experimental details and lack of controls. Both reviewers provided constructive comments that should allow the authors to prepare a paper for a specialty journal.

*Reviewer #1:*

In its current form, this paper is a blast from the past. I taught developmental biology in the late 1960's and this type of paper was the hot topic of the time. Now we have a paper listing a "component" of a hormonal extract that is not a peptide – but rather is methylfarnesoate, which suppresses maturation in the model worm.

The authors think that the identification of this molecule deserves a placement of the paper in *eLife*., but at this point I tend to disagree.

The authors state: “The identification of MF as a first molecular candidate for nereidin paves the way for more detailed analyses on how this factor is regulated, and how different hormone levels translate into different physiological responses. For instance, different hypotheses have been made concerning the extent of nereidin regulation: studies in Nereis have led to the concept that nereidin concentrations drop to zero when animals enter maturation (Durchon and Porchet 1970; Golding 1983). In contrast, earlier studies in *Platynereis* argued only for a decrease in nereidin secretion from the mature head (Hauenschild 1966). Our comparison of MF levels in premature and mature animals argues that the latter hypothesis is more likely to be correct, at least for *Platynereis”*.

This is basically a very good analytical biochemistry paper that focused on a developmental problem – but didn't provide any insight into how the molecule works or how it is regulated.

*Reviewer #2:*

The paper 'Discovery of methylfarnesoate as the annelid brain hormone reveals an ancient role of sesquiterpenoids in reproduction' is an intriguing read. The fact that a sesquiterpenoid can be part of the long sought for nereidin is a significant finding and would be of crucial importance to reinvestigate the effects of commonly used JH analogues as insecticides. The authors have evidently spent a great amount of time in setting up successful assays. Having said this, I have some major reservations about the methods and the conclusions before I am completely convinced.

1) Please adhere to the MIQE standards for publishing q-RT-PCR results. I am missing qPCR primer efficiencies, the Methods section does not tell me which controls were used -RT, NTC, the PCR cycle, there is no mention of a DNAse treatment, amount of technical replicates, does not mention how the stability of the chosen reference genes was tested, were these selected from a whole suite of reference genes, as is required for certain algorithms (geNorm, Normfinder…), if there is a transcriptome available, this should not involve that much extra work, does state that in case only one stable reference gene was used in some cases, MIQE standards require you to use at least two. Since most conclusions are based on the q-RT-PCR results, it is vital this is performed correctly. As the manuscript is written now, there is no way of knowing.

2) Some of the effects seem to be rather weak if the conclusion of the manuscript is to be that decreasing levels of MF are crucial for reproduction. Can the authors explain why a one-tailed t-test is performed? Since MF is actually crucial for vitellogenesis in insect with ancestral characteristics, I don't think the hypothesis can automatically be that MF will reduce Vg mRNA levels… For several of the experiments, the two-sided t-test does not show significant effects (Figure 2, Figure 3. Can the authors explain why ANOVA was performed followed by one-sided t-tests? This seems a little unconventional to me.

3) Weird that when I blast the Lottia Met ortholog in GenBank with insects as a parameter for organism search set, I only retrieve 'Circadian locomotor output cycles protein kaput' as hits. If this is a true ortholog of Met, I would expect upon re-blast to at least get Met hits? Also the *Platynereis* sequence accession number is not available, so I have no way of knowing if this sequence, which apparently clearly is related to the Lottia clock, is a true Met.

4) I find the relative expression level changes in Met when administering MF unexpected since to my knowledge there is little dynamic range in Met expression levels in Arthropods, in these animals, downstream genes, like Kr-h1 are way more affected by changing JH levels, this could be discussed more.

5) Differences in in vitro and in vivo effects between administration of methoprene is also surprising. Methoprene should be more stable, a bigger effect could be expected. Was the actual JHIII tested? This could be relevant to claiming bio-availability issues,

6) Figure 4: I'm missing the non-decapitated animals as a control here.

7) Figure 2: Can a methoprene and JH treatment be included in the data for Figure 2. I'm missing this as somewhat of a positive control.

Figure 2: Is the 0.1 nM treatment really necessary for the statement you want to make? Some values are higher than the control. If yes, maybe it would be advisable to get more resolution in this range.

8) Figure 5: Can you please state whether non-treated actually means treated with 0.01% DMSO? This should be the case, but is not clear from the legend. Please use at least two stable housekeeping genes.

---

## [Author Response]

[Editors’ note: the author responses to the first round of peer review follow.]

*The two reviewers had seemingly different specific concerns, but both clearly had reservations about the impact of the paper. The first reviewer was concerned that the identification of a signaling molecule, while interesting, is not sufficient for publication in eLife without some clear mechanistic understanding of how the molecular basis works. The second reviewer was concerned about experimental details and lack of controls. Both reviewers provided constructive comments that should allow the authors to prepare a paper for a specialty journal.*

*Reviewer #1:*

*In its current form, this paper is a blast from the past. I taught developmental biology in the late 1960's and this type of paper was the hot topic of the time. Now we have a paper listing a "component" of a hormonal extract that is not a peptide – but rather is methylfarnesoate, which suppresses maturation in the model worm.*

The authors think that the identification of this molecule deserves a placement of the paper in eLife, but at this point I tend to disagree.

*The authors state: “The identification of MF as a first molecular candidate for nereidin paves the way for more detailed analyses on how this factor is regulated, and how different hormone levels translate into different physiological responses. For instance, different hypotheses have been made concerning the extent of nereidin regulation: studies in Nereis have led to the concept that nereidin concentrations drop to zero when animals enter maturation (Durchon and Porchet 1970; Golding 1983). In contrast, earlier studies in Platynereis argued only for a decrease in nereidin secretion from the mature head (Hauenschild 1966). Our comparison of MF levels in premature and mature animals argues that the latter hypothesis is more likely to be correct, at least for Platynereis”. This is basically a very good analytical biochemistry paper that focused on a developmental problem – but didn't provide any insight into how the molecule works or how it is regulated.*

In line with the procedure agreed upon in discussion with the editors, we have revised the manuscript to improve the two aspects that the reviewer highlighted:

1) Mechanistic insight (“how the molecule works”). We significantly revised our Discussion (relevant sections: second and fourth paragraphs) to emphasize more clearly the significant novel mechanistic insight that this study achieved. This includes:

1.1) Identification of specific target cells.

We were not only able to assign overall effects of methylfarnesoate on the whole organism, but rather could pinpoint the impact of the hormone on a specific target cell type (the eleocyte). As we show, this cell type consistently also express the seqsquiterpenoid receptor orthologue of the bristleworm. Eleocytes are known to play pivotal roles in reproductive decision, and identifying them as direct targets of the brain hormone is significant insight into the mechanism how reproduction is controlled by the brain hormone.

1.2) The role of methylfarnesoate to directly regulate vitellogenesis.

We report that methylfarnesoate has a significant effect on the expression of vitellogenin, the yolk precursor, in eleocytes. As yolk makes up almost half of the biomass of a mature female, and is thought to be exclusively produced by eleocytes, the regulation we report is strong mechanistic evidence for a major and relevant metabolic switch being controlled by the brain hormone.

1.3) Methylfarnesoate autoregulates is putative receptor.

This insight results from quantification of receptor gene expression both in a context where endogenous levels of the hormone are sinking (Figure 4) and from the addition of increased amounts of the hormone (Figure 4). This is relevant mechanistic insight, as it provides a plausible mechanism for how the target cells are able to interpret a graded regulation of the hormone into a sharp biological response, as it is required for an all-or-nothing mode of reproduction.

2) Regulation of methylfarnesoate (“how the molecule […] is regulated”).

The regulation of methylfarnesaoate is one of the criteria that we explicitly address in the manuscript. We developed and validated a sensitive quantitative mass spectrometry assay exactly for this purpose, and assessed regulation of methylfarnesoate in the relevant period of development (see subsection “MF concentration drops in the course of maturation”, Figure 3 and corresponding supplements). In the revised version of the manuscript, this point is also explicitly covered in our discussion on the mechanistic insight provided by the manuscript (see above; Discussion, second paragraph), and the respective conceptual outlook (Discussion, third paragraph).

*Reviewer #2:*

*The paper 'Discovery of methylfarnesoate as the annelid brain hormone reveals an ancient role of sesquiterpenoids in reproduction' is an intriguing read. The fact that a sesquiterpenoid can be part of the long sought for nereidin is a significant finding and would be of crucial importance to reinvestigate the effects of commonly used JH analogues as insecticides. The authors have evidently spent a great amount of time in setting up successful assays. Having said this, I have some major reservations about the methods and the conclusions before I am completely convinced.*

We appreciate the careful review by this reviewer, and thank the reviewer for pointing out the significance of our discovery for invertebrate biology. We are confident that by addressing all of the raised questions (see below) and by adding the additional dataset on pyriproxyfen, we provide convincing support for all of our findings.

*1) Please adhere to the MIQE standards for publishing q-RT-PCR results. I am missing qPCR primer efficiencies, the Methods section does not tell me which controls were used -RT, NTC, the PCR cycle, there is no mention of a DNAse treatment, amount of technical replicates, does not mention how the stability of the chosen reference genes was tested, were these selected from a whole suite of reference genes, as is required for certain algorithms (geNorm, Normfinder…), if there is a transcriptome available, this should not involve that much extra work, does state that in case only one stable reference gene was used in some cases, MIQE standards require you to use at least two. Since most conclusions are based on the q-RT-PCR results, it is vital this is performed correctly. As the manuscript is written now, there is no way of knowing.*

1.1) Concerning the request to provide clearer technical details, these are now mentioned / expressed more clearly in the Methods section, (subsection “qRT-PCR analyses”, as well as in the applicable legends. In particular:

a) qPCR primer efficiencies are provided in Table 1 (respective calculations described in the aforementioned subsection, first paragraph);

b) qPCR cycles are described in the fourth paragraph;

c) qPCR controls are specified in lines the last two paragraphs;

d) DNAse treatments and reaction conditions are described in the second paragraph;

e) The details on technical replicates and further reaction details have been added to the Methods section as well (second and third paragraphs).

Concerning the use of reference genes, we revised the manuscript in the following way:

1.2) Choice of reference genes and overall quantification standards.

The Methods section now clearly outlines the criteria by which reference genes were chosen, and the fact that expression levels were generally controlled by the use of two reference genes, as requested by the reviewer (subsection “qRT-PCR analyses”, last two paragraphs).

The only exception to this two-reference-gene-rule, also addressed in the revised paper (subsection “qRT-PCR analyses”, fourth paragraph), were the qPCRs performed on coelomocytes in the context of the large biochemical fractionation assay. Here, only limited material was available for each fraction, which is why only a single, previously benchmarked reference gene (rps9, references: see below) was used for quantification. Additional data – supplied as Figure 2—figure supplement 2 (see below) demonstrate the stability of rps9 in these assays.

Performing these experiments on an even larger scale would not only require a large investment of time and reagents, but also require sacrificing hundreds of animals, in order to obtain suitable amounts of coelomocytes and brain hormone extracts. We do not consider this investment appropriate, especially as the qPCRs in this context did not primarily serve to make quantitative statements, but rather to determine active fractions (in a binary fashion), whereas the identified substance was then later validated in numerous separate, and well-controlled assays. In keeping with this argument, we have systematically avoided any quantitative statement from the respective section, and more precisely describe the “recovery” of bioactivity in the active fractions (subsection “Methylfarnesoate is the active component of the brain hormone and acts at physiological concentrations”).

1.3) Suitability of reference genes.

In the respective methods section (subsection “qRT-PCR analyses”, last two paragraphs), we explicitly refer the reader to previous publications in which two of these reference genes (rps9 and cdc5) have already been validated for their suitability in *Platynereis* qPCR assays (see e.g. Dray et al. (2010) Science 329(5989), 339–342; Zantke et al. (2013). Cell Reports, 5(1), 99–113; Tosches et al. (2014) Cell, 159(1), 46–57). In further support of the suitability of these genes, we also included a graph that shows consistent quantification of *vtg* against either of these two genes in our own assay (Figure 1—figure supplement 2).

In the revised version, we also include a similar, separate quantification for the cases where we used one of the additional reference genes *S-adenosylmethionine synthase (sams)* and *cation-idependent mannose-6-phosphate receptor (cim6pr)*. In either case – presented as Figure 3—figure supplement 4; Figure 4—figure supplement 1, respectively – the statements on significant gene regulation are fully supported by either evaluation, and the respective summary graphs are presented in the main figures. This is strong support for the suitability of these genes for the conclusions drawn from these assays.

Finally, as outlined above, the qPCRs performed on coelomocytes in the context of the large biochemical fractionation assay were only normalised against the single reference gene rps9. In this context, we assessed the stability of rps9 in cell culture by plotting the normalised Ct-values for each experimental series, demonstrating that – as expected from the published accounts – there is a high stability of these values (SD of 3%) (Figure 2—figure supplement 2).

Together, these clarifications and additional data all support the robustness of the performed qPCR assays, and also the validity of all relevant conclusions that we drew from these experiments.

*2) Some of the effects seem to be rather weak if the conclusion of the manuscript is to be that decreasing levels of MF are crucial for reproduction. Can the authors explain why a one-tailed t-test is performed?*

2.1) For testing for the ability of fractions to act as nereidin / brain hormone, we generally only performed one-sided t-tests, as brain hormone activity is defined as inhibiting maturation, and this is the hypothesis tested in the experiments. As we discover MF in the most active brain hormone fraction and then show that application of authentic MF lowers *vtg*-transcript abundance in separate, well-controlled analyses, we believe that a one-sided t-test is appropriate for the comparisons performed at this stage.

*Since MF is actually crucial for vitellogenesis in insect with ancestral characteristics, I don't think the hypothesis can automatically be that MF will reduce Vg mRNA levels…*

2.2) As to the comparison of our discoveries with insect data – and the question if these have predictive power – please see our common response below (point 4).

*For several of the experiments, the two-sided t-test does not show significant effects (Figure 2, Figure 3. Can the authors explain why ANOVA was performed followed by one-sided t-tests? This seems a little unconventional to me.*

2.3) We appreciate the effort of the reviewer to recalculate our statistics. As we stated in the Materials and methods section (subsection “Data analysis and statistics”), however, we performed Welch`s t-test, which is a t-test that does not assume equal variances (homoscedasticity) as does Student`s t-test; Welch`s t-test is therefore not prone to heteroscedasticity (https://en.wikipedia.org/wiki/Welch%27s_t-test). This makes Welch`s t-test better suited for unequal sample sizes (as they occur in our analyses) than Student`s t-test. Using Welch`s t-test yields significance in a two-sided test for Figure 2.

In Figure 2, only a one-sided t-test is performed, as we again are only interested in the inhibitory properties of MF and the comparison to PA and RA. However, the values obtained by R using a pairwise t-test are corrected for multiple testing, for all six combinations, out of which only the three referring to the control (control vs. MF, control vs. PA and control vs. RA) are relevant for our statement. If we perform multiple testing for only these three combinations, the p-values in a two sided t-test would be:

control vs. MF: 0.0419

control vs. PA: 0.3608

control vs. RA: 0.1907

For Figure 3, the reviewer is indeed correct in that a two sided t-test would be appropriate, and that the current data do not show statistical significance in such a test. In order to test our statement more rigorously, we now included more replicates in the data. In the revised analysis, we do reach statistical significance with a two-sided t-test (see Figure 3 and the corresponding source data). We additionally also corrected the statistical test from a one-sided t-test to a two-sided t-test in Figure 3 and Figure 4, where this test is also mandatory.

2.4) As to the last argument, the reviewer is right that first performing an ANOVA and then t-test seems unconventional. However, this does not affect the statistical statement. The reason why we did this is as follows. The classical approach for testing for group differences in more than one comparison is a one-way ANOVA. However, this only indicates that there are statistical differences (if any) between any of the groups tested. To resolve in which of the groups show statistical differences, to do this, classically a *post-hoc* test, in our case the pairwise t-test with correction for multiple testing, is performed.

3) Weird that when I blast the Lottia Met ortholog in GenBank with insects as a parameter for organism search set, I only retrieve 'Circadian locomotor output cycles protein kaput' as hits. If this is a true ortholog of Met, I would expect upon re-blast to at least get Met hits? Also the Platynereis sequence accession number is not available, so I have no way of knowing if this sequence, which apparently clearly is related to the Lottia clock, is a true Met.

3.1) As to the accession numbers for *Platynereis* sequences, these were already included in the first version of the manuscript (see legend Figure 3 / Methods section “gene identification and confirmation”). But following standard procedures, these sequences are not openly accessible until the publication has gone into print. In order to allow the reviewer to validate the analysis, we refer the reviewer to [Supplementary-material SD5-data].aln that contains the full alignment file for the phylogenetic tree of all displayed Met and Clock homologs.

3.2) As to the argument that blast may yield a different “closest” gene than the tree, this phenomenon is well known in molecular phylogeny, as the blast algorithm scores local similarities, whereas phylogeny is the dedicated method to assess the most likely evolutionary scenario. Depending on selection pressure, proteins or whole protein families evolve with different evolutionary speed, which is typically reflected by differences in branch length in the molecular phylogenetic analysis. As the reviewer will appreciate, both our original and our revised tree (Figure 3) shows much shorter branch lengths for Clock proteins, and longer ones for the Met group. Hence, a simple similarity search with a lophotrochozoan Met orthologue may indeed pick up an insect Clock protein (rather than an insect Met protein) as a first annotated hit, because the absolute distance (i.e sequence variation) appears shorter to Clock.

Please note that this situation will obviously change with the discovery – and proper annotation – of *Platynereis* Met and other lophotrochozoan Met orthologs, such as the *Lymnaea* candidate that we have added to the revised tree.

*4) I find the relative expression level changes in Met when administering MF unexpected since to my knowledge there is little dynamic range in Met expression levels in Arthropods, in these animals, downstream genes, like Kr-h1 are way more affected by changing JH levels, this could be discussed more.*

4) Here and in other sections (point 2: expectation for an inhibition of vitellogenesis; point 5 below), the reviewer correctly points out differences between the worms and several insect models.

While we appreciate the interest of these differences for discussion (and have improved these aspects accordingly – see below), we would like to emphasize clearly that this study did not start with a candidate approach, looking for the presence and activity of insect-type juvenile hormones in a lophotrochozoan model. Rather, we uncovered the identity of a long-sought hormone in a lophotrochozoan reference species, in which sesquiterpenoids are commonly thought not to be present, and then characterized this candidate and its effects further. Assuming that the split between ecdysozoans (the early arthropod ancestors) and lophotrochozoans occurred some 500 million years ago, the overall separation between insects and annelids is around a billion years. On the one hand, this makes the discovery of sesquiterpenoids in annelids such a relevant finding, but on the other hand, this clearly also has left ample time for modification of the ancient system in either lineage.

As a consequence of these facts, we do not see that our analysis should in any way be logically guided by the assumption that the role of the identified hormone should by default be equivalent to the role of sesquiterpenoids in insects or other arthropods, and we apologize should the previous version of the manuscript have lent support to that impression. Likewise, we believe that differences between the particular effects of sesquiterpenoids in either branch are perfectly possible, if not expected, and that studying these differences – and reconstruction commonalities as well as divergences – is an interesting task for future analyses, for which our study just paves the way.

To clarify these points better, our revised Discussion now explicitely refers to insects at three sections:

a) We explicitly mention that both similarities and differences exist in the role of sesquiterpenoid hormones between these systems when first discussing the function vitellogenesis (third paragraph);

b) In the fifth paragraph, we refer to the inferred ancestry of sesquiterpenoid (and steroid hormone) systems in the ancestor of arthropods and annelids;

c) In the sixth and seventh paragraphs, we highlight both similarities and differences between the systems in our discussion on the possible ancient function of the sesquiterpenoid system.

*5) Differences in in vitro and in vivo effects between administration of methoprene is also surprising. Methoprene should be more stable, a bigger effect could be expected. Was the actual JHIII tested? This could be relevant to claiming bio-availability issues.*

5) We thank the reviewer for pointing out these aspects. As to the role of insect juvenile hormones (like JHIII), for the reason just outlined, we consider this test less relevant for the main discovery that we report, but rather relevant for more detailed studies on the degree of divergence of the sesquiterpenoid system – in both arthropods and annelids – from their common ancestor. As a technical side note, we have not been able to obtain pure preparations of distinct insect juvenile hormones to be able to test this in our bioassay. We have, however, introduced a clarifying statement on cross-reactivities in the section where we discuss classical reports on the bioactivity of annelid extracts in insect juvenile hormone bioassays (Discussion, sixth paragraph).

Concerning the expectancy for the effects of methoprene – or JHIII – on whole worms, one factor that may be relevant is the efficiency by which these substances can diffuse into / are taken up by the animals. There is a variety of sesquiterpenoid-based insect growth regulators with different chemical side groups, and we note that MF is more lipophilic than methoprene, which may explain why methoprene has more subtle effects at the tested conditions. However, we also note that there are additonal parameters (time, temperature, light regimes) that may play into the availability and efficacy of these drugs.

As we are unable to provide a full series of conditions, we decided that for the revised version, we would leave out more extensive speculations about the bioavailability, and rather focus on the immediate effects we observed in the bioassay. Moreover, we included the dataset on pyriproxyfen, which is a chemically distinct, but highly used insecticide, and shows clear activity in the bioassay as well. This finding further substantiates similarities between the signalling pathways in insects and annelids, and also supports the relevance of our evolutionary discovery for the specificity of pest control mechanisms.

*6) Figure 4: I'm missing the non-decapitated animals as a control here.*

As the text details, the maturation is synchronized by decapitation, which is why non-decapitated animals would not be suitable as controls (subsection “Methylfarnesoate exerts a dominant repressive effect on vitellogenesis in vivo, and worms are susceptible to insecticides targeting the pathway”, second paragraph). However, we revised this section and the corresponding legend to point out more clearly that the control animals were also appropriately treated with DMSO.

*7) Figure 2: Can a methoprene and JH treatment be included in the data for Figure 2. I'm missing this as somewhat of a positive control.*

7.1) For the reasons outlined under point (4), the effect of insect Juvenile hormones like JHIII refers to the sepcific question to which extent sesquiterpenoid hormone ligands have exchangeable functions between these two – evolutionarily distant – clades.

For the given reasons (scope of this study, logical flow of the arguments, inavailability of pure fractions of JHIII), we can at present not include the dataset the reviewer suggests, but also do not think that it will be relevant for supporting the main findings of our study.

Figure 2: Is the 0.1 nM treatment really necessary for the statement you want to make? Some values are higher than the control. If yes, maybe it would be advisable to get more resolution in this range.

7.2) This test provided first evidence that MF acts at levels that are considered physiological for a hormone (1-100nM); they also match with the levels that were later determined to be present in the head. In that sense, to discover that the lower bound of significant effects is at around 1nM is neither unexpected nor contradictory, nor does it, in our eyes, require a deeper analysis in that range.

We understand that the reviewer may take the enlarged spread of data in this panel (in the positive range) as supportive evidence that low levels of sesquiterpenoids may be involved in positive regulation of vitellogenesis (see point 2.2 above). While we cannot exclude this, the current data do not provide statistically robust evidence for this.

As a result, we propose to leave the figure panel as is, but we more explicitly discuss the possibility of low-level-requirements of MF for vitellogenesis in the revised Discussion section (third and seventh paragraphs).

*8) Figure 5: Can you please state whether non-treated actually means treated with 0.01% DMSO? This should be the case, but is not clear from the legend. Please use at least two stable housekeeping genes.*

8) Concerning the relevance of using DMSO-treated animals as controls: The reviewer is of course right, and while this was described in the Methods section already, we have now added this clarification at several places in the manuscript and figure legends.

As to the argument of housekeeping genes, please see the more detailed response to the qRT-PCR experiments above (section 1.2).